# The zebrafish mutant *dreammist* implicates sodium homeostasis in sleep regulation

Ida L Barlow[1†], Eirinn Mackay[1‡], Emily Wheater[1§], Aimee Goel[1], Sumi Lim[1], Steve Zimmerman[2], Ian Woods[3], David A Prober[4], Jason Rihel[1*]

[1]Department of Cell and Developmental Biology, University College London, London, United Kingdom; [2]Department of Molecular and Cellular Biology, Harvard University, Cambridge, United States; [3]Ithaca College, New York, United States; [4]Division of Biology and Biological Engineering, California Institute of Technology, Pasadena, United States

*For correspondence:
j.rihel@ucl.ac.uk

Present address: [†]MRC London Institute for Medical Sciences, Imperial College, London, United Kingdom; [‡]Sainsbury Wellcome Centre for Neural Circuits and Behaviour, University College London, London, United Kingdom; [§]MRC Centre for Reproductive Health, University of Edinburgh, Edinburgh, United Kingdom

Competing interest: The authors declare that no competing interests exist.

**Abstract** Sleep is a nearly universal feature of animal behaviour, yet many of the molecular, genetic, and neuronal substrates that orchestrate sleep/wake transitions lie undiscovered. Employing a viral insertion sleep screen in larval zebrafish, we identified a novel gene, *dreammist* (*dmist*), whose loss results in behavioural hyperactivity and reduced sleep at night. The neuronally expressed *dmist* gene is conserved across vertebrates and encodes a small single-pass transmembrane protein that is structurally similar to the $Na^+,K^+$-ATPase regulator, FXYD1/Phospholemman. Disruption of either *fxyd1* or *atp1a3a*, a $Na^+,K^+$-ATPase alpha-3 subunit associated with several heritable movement disorders in humans, led to decreased night-time sleep. Since *atpa1a3a* and *dmist* mutants have elevated intracellular $Na^+$ levels and non-additive effects on sleep amount at night, we propose that Dmist-dependent enhancement of $Na^+$ pump function modulates neuronal excitability to maintain normal sleep behaviour.

## eLife assessment

This study offers new **fundamental** information on a role for the sodium/potassium pump in sleep regulation. Elegant methods were used to provide **compelling** evidence supporting the claim. The work will be of interest to sleep researchers in zebrafish as well as in other species for future investigation.

## Introduction

The ability of animals to switch between behaviourally alert and quiescent states is conserved across the animal kingdom (*Cirelli, 2009*; *Joiner, 2016*). Fundamental processes that govern the regulation of sleep-like states are shared across species, such as the roles of circadian and homeostatic cues in regulating the time and amount of sleep, stereotyped postures, heightened arousal thresholds, and the rapid reversibility to a more alert state (*Joiner, 2016*). The near ubiquity of sleep implies that it serves ancient functions and is subject to conserved regulatory processes. However, many key molecular components that modulate sleep and wake states remain undiscovered.

Over the past two decades, investigations into sleep and arousal states of genetically tractable model organisms, such as *Drosophila melanogaster*, *Caenorhabditis elegans*, and *Danio rerio* (zebrafish), have uncovered novel molecular and neuronal components of sleep regulation through gain- and loss-of-function genetic screens (reviewed in *Barlow and Rihel, 2017*; *Sehgal and Mignot,*

*2011*). The power of screening approaches is perhaps best exemplified by the first forward genetic sleep screen, which identified the potassium channel *shaker* as a critical sleep regulator in *Drosophila* (*Cirelli et al., 2005*). This result continues to have a lasting impact on the field as not only did subsequent sleep screening efforts uncover the novel Shaker regulator *sleepless* (*Koh et al., 2008*), but investigations into Shaker's beta subunit Hyperkinetic ultimately revealed a critical role for this redox sensor linking metabolic function to sleep (*Bushey et al., 2007*; *Kempf et al., 2019*).

Disparate screening strategies across model organisms continue to unveil novel sleep modulators in both invertebrate and vertebrate model systems. For example, the roles of RFamide receptor DMSR-1 in stress-induced sleep in *C. elegans* (*Iannacone et al., 2017*) and SIK3 kinase in modulating sleep homeostasis in mice (*Funato et al., 2016*) were identified in genetic screens. Moreover, a gain-of-function screening strategy in *Drosophila* revealed the novel sleep and immune regulator, *nemuri* (*Toda et al., 2019*), and a zebrafish overexpression screen uncovered the secreted neuropeptides neuromedin U and neuropeptide Y, which decrease and increase sleep, respectively (*Chiu et al., 2016*; *Singh et al., 2017*). The success of screening strategies in revealing novel sleep-wake regulatory genes suggests that more sleep signals likely remain to be discovered.

One of the lessons from these genetic screens is that many of the uncovered genes play conserved roles across species. For example, Shaker also regulates mammalian sleep (*Douglas et al., 2007*) and RFamides induce sleep in worms, flies, and vertebrates (*Lee et al., 2017*; *Lenz et al., 2015*). Nevertheless, not every invertebrate sleep-regulatory gene has a clear vertebrate homolog, while some human sleep/wake regulators, such as the narcolepsy-associated neuropeptide hypocretin/orexin (*Chemelli et al., 1999*; *Lin et al., 1999*; *Peyron et al., 2000*; *Sakurai, 2013*), lack invertebrate orthologs. Therefore, genetic sleep screens in vertebrates are likely to provide added value in uncovering additional regulatory components required to control the initiation and amount of sleep in humans.

While sleep screening in mammals is feasible (*Funato et al., 2016*), it remains an expensive and technically challenging endeavour. With its genetic tractability, availability of high-throughput sleep assays (*Rihel and Schier, 2013*), and conserved sleep genetics, such as the hypocretin, melatonin, locus coeruleus, and raphe systems (*Gandhi et al., 2015*; *Singh et al., 2015*; *Oikonomou et al., 2019*; *Prober et al., 2006*), the larval zebrafish is an attractive vertebrate system for sleep screens. We took advantage of a collection of zebrafish lines that harbour viral insertions in >3500 genes (*Varshney et al., 2013*) to perform a targeted genetic screen. We identified a short-sleeping mutant, *dreammist*, with a disrupted novel, highly conserved vertebrate gene that encodes a small single-pass transmembrane protein. Sequence and structural homology to the Na$^+$/K$^+$ pump regulator FXYD1/Phospholemman suggests that Dreammist is a neuronal-expressed member of a class of sodium pump modulators that is important for regulating sleep-wake behaviour.

## Results

### Reverse genetic screen identifies *dreammist*, a mutant with decreased sleep

We used the 'Zenemark' viral insertion-based zebrafish gene knock-out resource (*Varshney et al., 2013*) to perform a reverse genetic screen to identify novel vertebrate sleep genes. This screening strategy offers several advantages compared to traditional chemical mutagenesis-based forward genetic screening approaches. First, unlike chemical mutagenesis, which introduces mutations randomly, viral insertions tend to target the 5′ end of genes, typically causing genetic loss of function (*Sivasubbu et al., 2007*). Second, because the virus sequence is known, it is straightforward to map and identify the causative gene in mutant animals. Finally, since viral insertions in the Zenemark collection are already mapped and sequenced, animals harbouring insertions within specific gene classes can be selected for testing (*Figure 1—figure supplement 1A*). This allowed us to prioritise screening of genes encoding protein classes that are often linked to behaviour, such as G-protein-coupled receptors, neuropeptide ligands, ion channels, and transporters (*Figure 1—source data 1*).

For screening, we identified zebrafish sperm samples from the Zenemark collection (*Varshney et al., 2013*) that harboured viral insertions in genes of interest and used these samples for in vitro fertilisation and the establishment of F2 families, which we were able to obtain for 26 lines. For each viral insertion line, clutches from heterozygous F2 in-crosses were raised to 5 days post-fertilisation (dpf) and tracked using videography (*Figure 1—figure supplement 1A*) to quantify the number and

duration of sleep bouts (defined in zebrafish larvae as inactivity lasting 1 minute or longer; *Prober et al., 2006*) and waking activity (time spent moving per active bout) over 48 hr. The genotypes of individual larvae were determined by PCR after behavioural tracking, with each larva assigned as wild type, heterozygous, or homozygous for a given viral insertion to assess the effect of genotype on sleep/wake behaviour. While most screened heterozygous and homozygous lines had minimal effects on sleep-wake behavioural parameters (*Figure 1—figure supplement 1B and C*), one homozygous viral insertion line, *10543/10543,* had a reduction in daytime sleep (*Figure 1—figure supplement 1B*) and an increase in daytime waking activity (*Figure 1—figure supplement 1C*) relative to their wild-type sibling controls. We renamed this *10543* viral insertion line *dreammist* (*dmist*).

In follow-up studies, we observed that animals homozygous for the viral insertion at this locus (*dmist$^{vir/vir}$*) showed a decrease in sleep during the day and a trend to sleep less at night compared to their wild-type siblings (*dmist$^{+/+}$*) (*Figure 1A*). *dmist* mutants had an almost 50% reduction in the average amount of daytime sleep (*Figure 1C*) due to a decrease in the number of sleep bouts (*Figure 1D*), whereas the sleep bout length at night was significantly reduced (*Figure 1E*). *dmist$^{vir/vir}$* larvae also exhibited significantly increased daytime waking activity, which is the locomotor activity while awake (*Figure 1B and F*). Because Zenemark lines can contain more than one viral insertion (17.6% of lines have ≥2 insertions; *Varshney et al., 2013*), we outcrossed *dmist$^{vir/+}$* fish to wild-type fish of the AB-TL background and retested *dmist* mutant fish over several generations. Normalising all the behavioural parameters to *dmist$^{+/+}$* controls with a linear mixed effects (LME) model showed consistent sleep changes in *dmist$^{vir/vir}$* fish over five independent experiments (*Figure 1G*). The *dmist$^{vir/vir}$* larvae consistently show a >50% decrease in sleep during the day due to a significant reduction in the number and duration of sleep bouts, as well as a large increase in waking activity (*Figure 1G*). The *dmist$^{vir/vir}$* mutants also had a significant reduction in sleep at night compared to wild-type siblings (*Figure 1G*). These effects on sleep and wakefulness are not due to alterations in circadian rhythms as behavioural period length in fish that were entrained and then shifted to free-running constant dark conditions was unaffected in *dmist$^{vir/vir}$* compared to wild-type sibling larvae (*Figure 1—figure supplement 2A–C*).

## The *dmist* gene encodes a novel, small transmembrane protein

Having identified a sleep mutant, we next sought to investigate the target gene disrupted by the viral insertion. Line *10543* (*dmist$^{vir}$*) was initially selected for screening due to a predicted disruption of a gene encoding a serotonin transporter (*slc6a4b*) on chromosome 5. However, mapping of the *dmist* viral insertion site by inverse-PCR and sequencing revealed that the virus was instead inserted into the intron of a small two-exon gene annotated in the Zv6 genome assembly as a long intergenic non-coding RNA (lincRNA; gene transcript ENSDART00000148146, gene name *si:dkey234h16.7*), which lies approximately 6 kilobases (kb) downstream of the *slc6a4b* gene in zebrafish. At least part of this region is syntenic across vertebrates, with a small two-exon gene identified adjacent to the genes *ankrd13a* and *GIT* in several vertebrates, including human and mouse (*Figure 2A*). Amplifying both 5′ and 3′ ends of zebrafish *si:dkey234h16.7* and mouse E13.5 *1500011B03-001* transcripts with Rapid Amplification of cDNA ends (RACE) confirmed the annotated zebrafish and mouse transcripts and identified two variants with 3′ untranslated regions (3′UTR) of different lengths in zebrafish (*Figure 2—figure supplement 1B*). To test whether the viral insertion in *dmist$^{vir/vir}$* disrupts expression of *si:dkey234h16.7* or neighbouring genes, we performed quantitative analysis of gene transcript levels in wild type and mutant *dmist* larvae by RT-qPCR. This revealed that the *dmist* viral insertion caused a >70% reduction in the expression of *si:dkey234h16.7* while the expression of the most proximal 5′ or 3′ flanking genes, *slc6a4b_Dr* and *ankrd13a_Dr*, were unaffected (*Figure 2B*, *Figure 2—figure supplement 1A*). Since this reduced expression is most consistent with *si:dkey234h16.7* being the causal lesion of the *dmist* mutant sleep phenotype, we renamed this gene *dreammist* (*dmist*).

Computational predictions indicated that the *dmist* transcripts contain a small open-reading frame (ORF) encoding a protein of 70 amino acids (aa) (*Figure 2C*). Querying the human and vertebrate protein databases by BLASTp using the C-terminal protein sequence of Dmist identified orthologs in most vertebrate clades, including other species of teleost fish, birds, amphibians, and mammals (*Figure 2A, C*). All identified orthologues encoded predicted proteins with an N-terminal signal peptide sequence and a C-terminal transmembrane domain (*Figure 2C*). The peptide sequence identity across orthologs ranged from 38 to 84%, with three peptide motifs (QNLV, CVYKP, RRR) showing high conservation across all vertebrates and high similarity for many additional residues (*Figure 2C*,

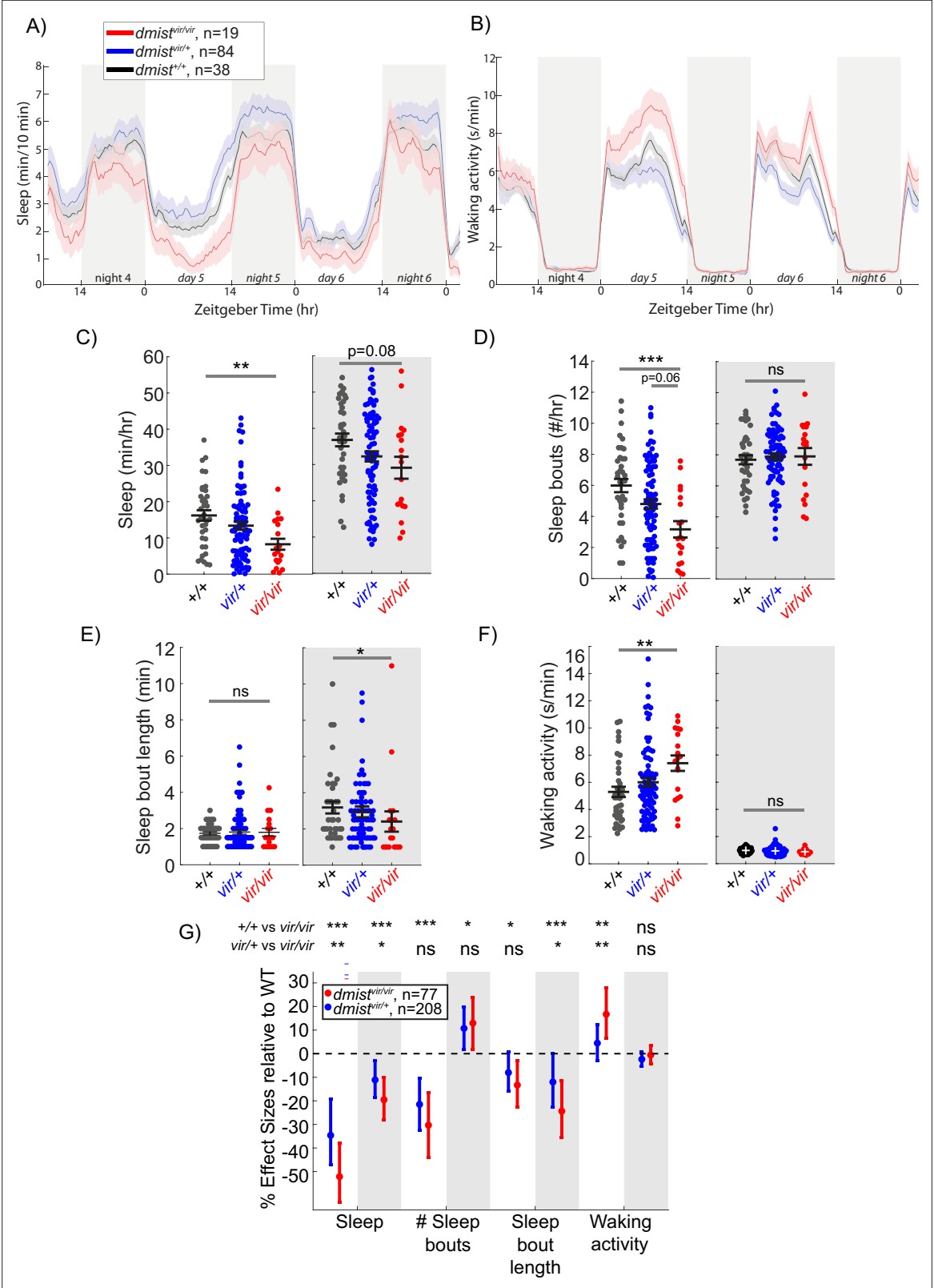

**Figure 1.** A viral insertion mini-screen identifies a short-sleeping mutant, *dreammist*. (**A, B**) Mean ± SEM sleep (**A**) and waking activity (**B**) of progeny from *dmist*^*vir/+*^ in-cross from original screen. White blocks show day (lights on) and grey blocks show night (lights off). Data is combined from two independent experiments. n indicates the number of animals. (**C–F**) Analysis of sleep/wake architecture for the data shown in (**A, B**). (**C**) Quantification of total sleep across 2 d and nights shows decreased day and night sleep in *dmist*^*vir/vir*^. Analysis of sleep architecture reveals fewer sleep bouts during

*Figure 1 continued on next page*

Figure 1 continued

the day (**D**) and shorter sleep bouts at night (**E**) in *dmist*<sup>vir/vir</sup> compared with sibling controls. (**F**) Daytime waking activity is also increased in *dmist*<sup>vir/vir</sup>. The black lines show the mean ± SEM, except in (**E**), which labels the median ± SEM. *p<0.05, **p<0.01, ***p<0.001; ns p>0.05; one-way ANOVA, Tukey's post hoc test. (**G**) Combining five independent experiments using a linear mixed effects model with genotype as a fixed effect and experiment as a random effect reveals *dmist*<sup>vir/vir</sup> larvae have decreased total sleep and changes to sleep architecture during both the day and night compared to *dmist*<sup>+/+</sup> siblings. Plotted are the genotype effect sizes (95% confidence interval) for each parameter relative to wild type. Shading indicates day (white) and night (grey). p-Values are assigned by an *F*-test on the fixed effects coefficients from the linear mixed effects model. *p<0.05, **p<0.01, ***p<0.001, ns p>0.05. n indicates the number of animals.

The online version of this article includes the following source data and figure supplement(s) for figure 1:

**Figure supplement 1.** A viral insertion screen for sleep-wake regulators.

**Figure supplement 2.** *dmist*<sup>vir/vir</sup> fish are hyperactive and have normal circadian rhythms.

**Source data 1.** Gene selection for screening.

*Figure 2—figure supplement 1D*). Additional searches by tBLASTn failed to identify any non-vertebrate *dmist* orthologs. In summary, we found that the *dreammist* gene, the expression of which is disrupted in *dmist*<sup>vir/vir</sup> fish with sleep phenotypes, encodes a protein of uncharacterised function that is highly conserved across vertebrates at both the genomic and molecular levels.

## Genetic molecular analysis of *dmist* expression in zebrafish and mouse

Because the viral insertion disrupts *dmist* throughout the animal's lifetime, we examined both the developmental and spatial expression of *dmist* to assess when and where its function may be required for normal sleep. Using the full-length transcript as a probe (*Figure 2—figure supplement 1B*), we performed in situ hybridisation across embryonic and larval zebrafish development. Maternally deposited *dmist* was detected in early embryos (two-cell stage) prior to the maternal to zygotic transition (*Giraldez et al., 2006*; *Figure 2D*). Consistent with maternal deposition of *dmist* transcripts, inspection of the 3' end of the *dmist* gene revealed a cytoplasmic polyadenylation element ('TTTTTTAT') that is required for zygotic translation of maternal transcripts (*Villalba et al., 2011*). At 24 hpf, transcripts were detected in regions that form the embryonic brain, such as ventral telencephalon, diencephalon, and cerebellum, and in the developing eye (*Figure 2D*). By 5 dpf, *dmist* transcripts were detected throughout the brain (*Figure 2D*). To test whether *dmist* transcripts are under circadian regulation, we performed RT-qPCR in fish that were entrained and then shifted to free-running constant dark conditions. In contrast with the robust 24 hr rhythmic transcription of the circadian clock gene *per1*, we did not detect any changes in *dmist* expression throughout the 24 hr circadian cycle (*Figure 1—figure supplement 2D*).

Consistent with brain expression in larval zebrafish, we identified the expression of *Dmist_Mm* in a published RNAseq dataset of six isolated cell types from mouse cortex (*Zhang et al., 2014*). We confirmed that *Dmist_Mm* is specifically enriched in neurons by hierarchical clustering of all 16,991 expressed transcripts across all six cells types, which demonstrated that *Dmist_Mm* co-clusters with neuronal genes (*Figure 2—figure supplement 1E*). Pearson correlation of *Dmist_Mm* with canonical markers for the six cell types showed that *Dmist_Mm* expression is highly correlated with other neuronal genes but not genes associated with microglia, oligodendrocytes, or endothelia. This result indicates that *dmist* is specifically expressed in neurons in both zebrafish and mouse (*Figure 2—figure supplement 1F*).

## Dmist localises to the plasma membrane

Although the *dmist* gene encodes a conserved ORF with a predicted signal peptide sequence and transmembrane domain (*Figure 2C*, *Figure 2—figure supplement 1G–I*), we wanted to confirm this small peptide can localise to the membrane, and if so, on which cellular compartments. To test these computational predictions, we transiently co-expressed GFP-tagged Dmist (C-terminal fusion) with a marker for the plasma membrane (myr-Cherry) in zebrafish embryos. Imaging at 90% epiboly revealed Dmist-GFP localised to the plasma membrane (*Figure 2E*). Conversely, introducing a point mutation into Dmist's signal peptide cleavage site (DmistA22W-GFP) prevented Dmist from trafficking to the plasma membrane, with likely retention in the endoplasmic reticulum (*Figure 2F*). Together, these

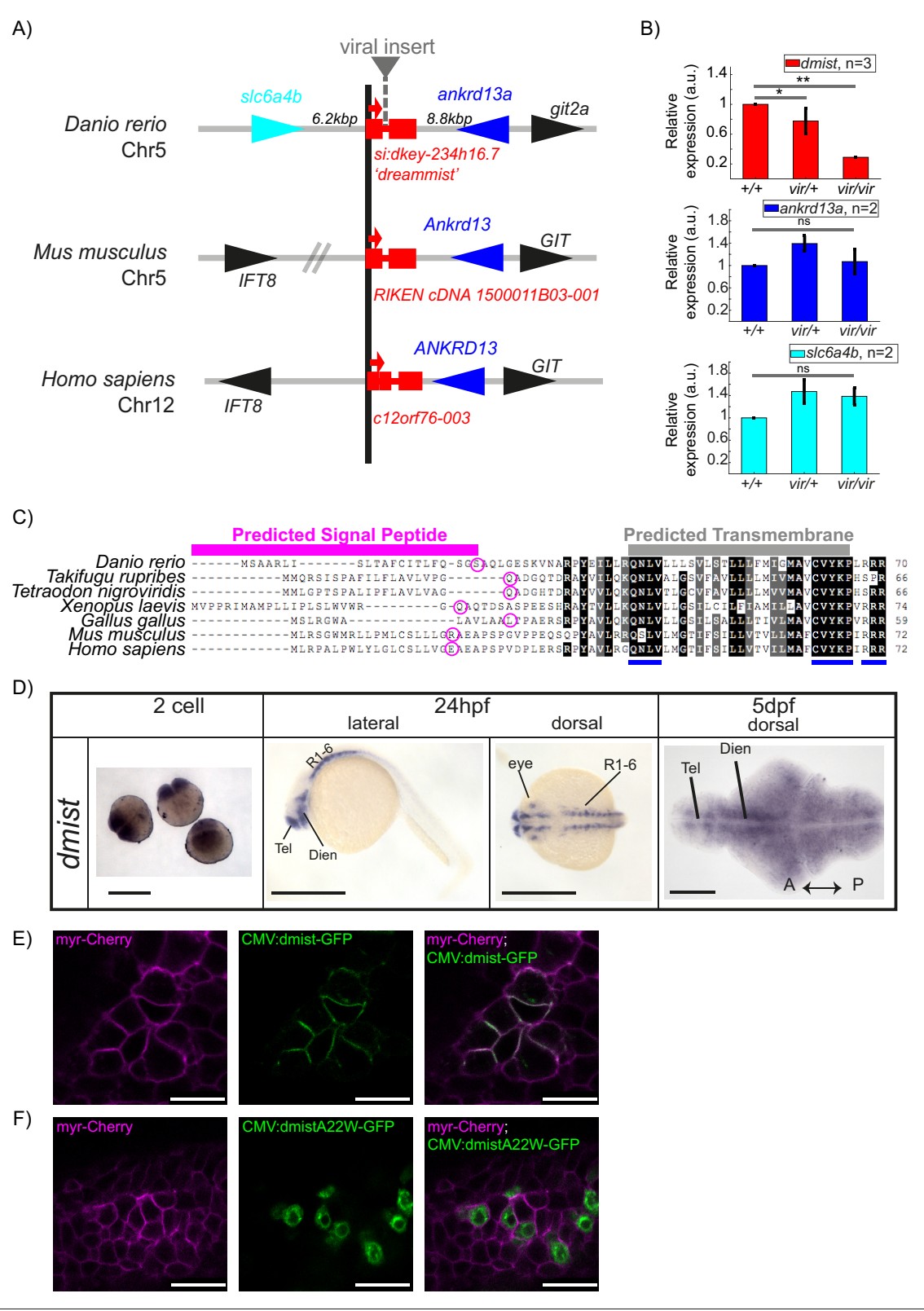

**Figure 2.** *dmist* encodes a conserved vertebrate single-pass transmembrane protein. (**A**) *dmist* mutants harbour a viral insertion in the first intron of *si:key-234h16.7*. *dmist* is syntenic with *Ankrd13* and *GIT* orthologs in mouse, human, and zebrafish. (**B**) RT-qPCR of *dmist* (red) show reduced expression of *dmist* and not the 5′ and 3′ flanking zebrafish genes, *slc6a4b* (cyan) and *ankrd13a* (blue), in *dmist*^vir/vir^ larvae compared to *dmist*^vir/+^ and *dmist*^+/+^ siblings. **p<0.01, *p<0.05; ns p>0.05; one-way ANOVA, Tukey's post hoc test. Data shows mean ± SEM normalised to the wild-type mean. (**C**) *dmist_Dr*

*Figure 2 continued*

contains an open-reading frame encoding a 70 amino acid protein that is conserved across vertebrates. All identified homologues have a predicted signal peptide sequence (magenta line), signal peptide cleavage site (magenta circle), and predicted transmembrane domain (grey), with additional highly conserved C-terminal motifs (blue lines). Identical amino acids in all species are shown in black; similar amino acids (80–99% conserved across species) are shown in grey. (D) In situ hybridisation using a *dmist* antisense probe reveals *dmist* is maternally deposited as it is detected at the two-cell stage. At 24 hpf, expression is restricted to regions containing neuronal precursors, and at 5 dpf expression is widespread throughout the brain. Tel, telencephalon; Dien, diencephalon; R1-6, rhombomeres 1–6; A, anterior; P, posterior. Scale bars = 0.5 mm (two-cell and 24 hpf), 0.1 mm (5 dpf). (E, F) Representative confocal image of 90% epiboly embryo co-injected at the one-cell stage with mRNA encoding membrane-RFP (magenta) and a plasmid encoding either C-terminal tagged Dmist-GFP (E, green) or DmistA22W-GFP (F, green). Scale bar = 25 μm.

The online version of this article includes the following figure supplement(s) for figure 2:

**Figure supplement 1.** *dmist* is enriched in neurons and requires the signal peptide cleavage site for membrane localisation.

data indicate that Dmist localises to the plasma membrane despite its small size, as computationally predicted.

## CRISPR/Cas9-generated *dmist*^i8^ mutant exhibits decreased night-time sleep

*dmist* expression was reduced by 70% in the viral insertion line, suggesting that *dmist*^vir^ is a hypomorphic allele. To confirm that the sleep phenotypes observed in *dmist*^vir/vir^ animals are due to the loss of Dmist function, we used CRISPR/Cas9 to create an independent *dmist* loss-of-function allele. We generated a zebrafish line in which the *dmist* gene contains an 8 bp insertion that causes a frameshift and early stop codon (*dmist*^i8^, **Figure 3A**). The *dmist*^i8^ allele is predicted to encode a truncated protein lacking the complete signal peptide sequence and transmembrane domain (**Figure 3B**), indicating that this is likely a null allele. RT-qPCR showed that *dmist* transcript levels were 60% lower in *dmist*^i8/i8^ fish compared to wild-type siblings, consistent with nonsense-mediated decay (**Figure 3—figure supplement 1A and B**; *Wittkopp et al., 2009*).

We next assessed the sleep and activity patterns of *dmist*^i8/i8^ fish. As seen in exemplar individual tracking experiments, *dmist*^i8/i8^ larvae sleep less at night due to fewer sleep bouts and also show an increase in waking activity relative to wild type and heterozygous mutant siblings (**Figure 3C–H**). This significant night-time reduction in sleep and increase in hyperactivity is also apparent when combining five independent experiments with a LME model to normalise behaviour across datasets (**Figure 3I**). Although *dmist*^vir/vir^ larvae also sleep less at night (**Figure 1G**), the large daytime reduction in sleep observed in *dmist*^vir/vir^ larvae is absent in *dmist*^i8/i8^ animals, perhaps due to the differences in genetic background that affect behaviour. Because the *dmist*^vir^ is likely a hypomorphic allele, we focused subsequent experiments on the CRISPR-generated *dmist*^i8/i8^ larvae.

To test whether the increased night-time activity of *dmist*^i8/i8^ mutants persists in older animals, we raised *dmist*^i8/i8^ mutants with their heterozygous and wild-type siblings to adulthood in the same tank and tracked individual behaviour for several days on a 14hr:10hr light:dark cycle. As in larval stages, *dmist*^i8/i8^ adults were hyperactive relative to both *dmist*^i8/+^ and *dmist*^+/+^ siblings, maintaining a higher mean speed at night (**Figure 3J–L**). This suggests that either Dmist affects a sleep/wake regulatory circuit during development that is permanently altered in *dmist* mutants or that Dmist is continuously required to maintain normal levels of night-time locomotor activity.

## Dmist is distantly related to the Na⁺/K⁺ pump regulator Fxyd1 (Phospholemman)

Because Dmist is a small, single-pass transmembrane domain protein without any clear functional motifs and has not been functionally characterised in any species, we searched for similar peptides that might provide clues for how Dmist regulates behaviour. Using the multiple sequence alignment tool MAFFT to align the zebrafish, mouse, and human Dmist peptides (*Katoh and Toh, 2010*) and seeding a hidden Markov model iterative search (JackHMMR) of the UniProt database (*Johnson et al., 2010*), we found distant homology between Dmist and Fxyd1/Phospholemman (**Figure 4A**), a small transmembrane domain peptide that regulates ion channels and pumps, including the Na⁺,K⁺-ATPase pump (*Crambert et al., 2002*). Dmist and Fxyd1 share 27–34% amino acid homology, including an RRR motif at the C-terminal end, although Dmist lacks a canonical FXYD sequence (**Figure 4A**). In addition, computational predictions using the AlphaFold protein structure database revealed structural

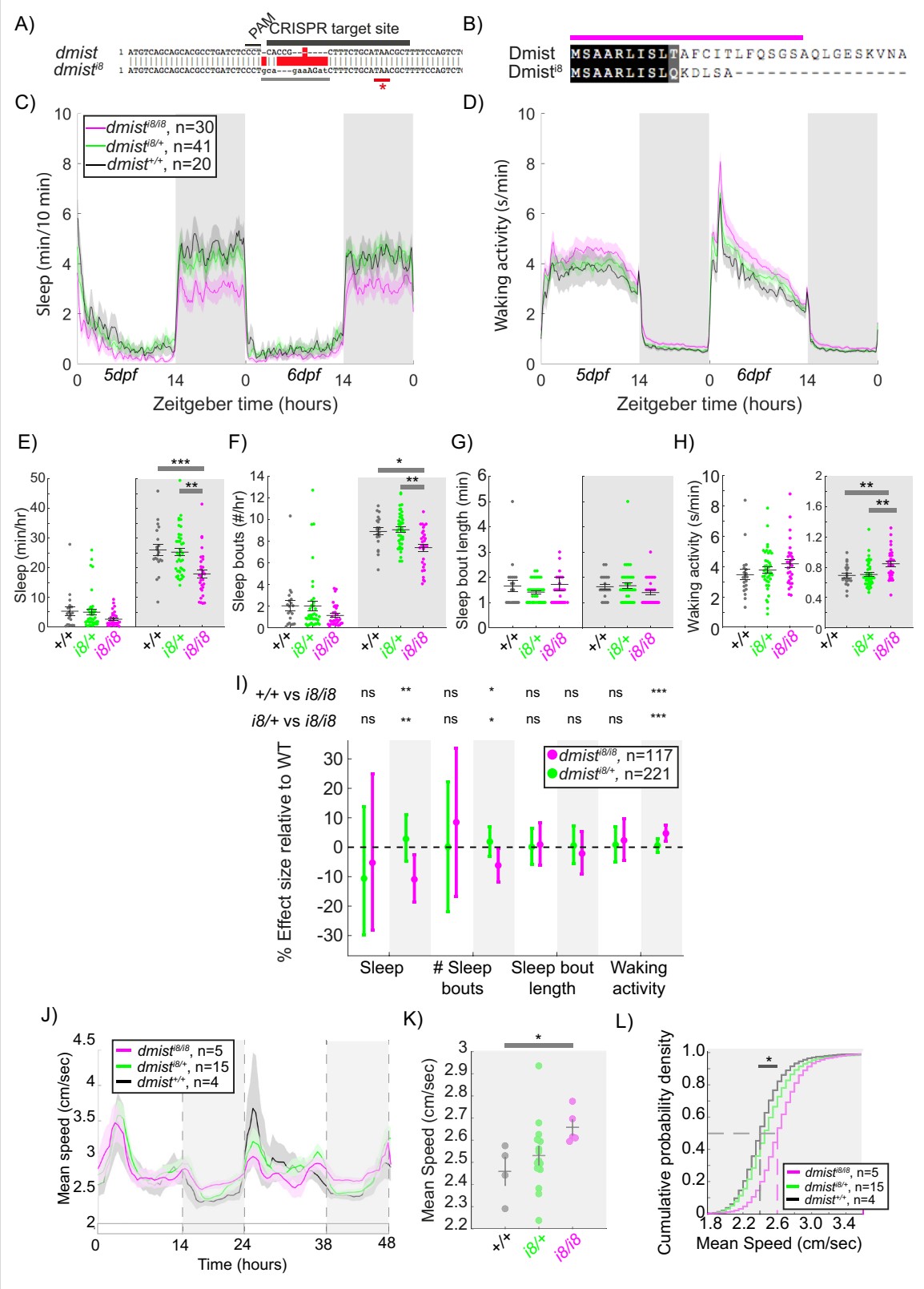

**Figure 3.** CRISPR-generated *dmist* mutants sleep less and are hyperactive at night. (**A**) CRISPR/Cas9 targeting of the first exon of *dmist* resulted in an 8 bp insertion (*dmist^i8*) (grey line) within the coding sequence, leading to an early stop codon (red line with *). Guide RNA target sequence and PAM sequence are shown as black bars. The sequence that is deleted in the mutant is indicated with a red bar. (**B**) Predicted Dmist^i8 peptide sequence lacks most of the N-terminal signal peptide sequence (magenta) and the full C-terminus. (**C, D**) Representative 48 hr traces of mean ± SEM sleep (**C**) and

*Figure 3 continued on next page*

Figure 3 continued

waking activity (D) shows decreased sleep and increased waking activity at night for *dmist*[i8/i8] fish compared to *dmist*[i8/+] and *dmist*[+/+] siblings. n = number of fish. (E–H) Analysis of sleep/wake architecture of the experiment depicted in (C, D) indicates that *dmist*[i8/i8] larvae sleep less at night (E) due to fewer sleep bouts (F). Sleep bout length is unchanged (G). Waking activity is also increased in *dmist*[i8/i8] fish (H). The black line represents the mean ± SEM except for (G), which is the median ± SEM. *p<0.05, **p<0.01, ***p<0.001; one-way ANOVA, Tukey's post hoc test. (I) Combining five independent experiments with a linear mixed effects model reveals *dmist*[i8/i8] fish sleep less at night due to fewer sleep bouts and also show increased waking activity at night. Plotted are the genotype effect sizes (95% confidence interval) for each parameter relative to wild type. Shading indicates day (white) and night (grey). p-Values are assigned by an *F*-test on the fixed effects coefficients from the linear mixed effects model. *p<0.05, **p<0.01, ***p<0.001, ns p>0.05. (J) Adult *dmist*[i8/i8] fish have a higher mean swim speed compared to their wild-type siblings at night. Data in (J) is quantified at night in (K). (J, K) show mean ± SEM. *p<0.05, one-way ANOVA. (L) Cumulative probability distribution of all night-time swim bout speeds in adult fish. The dashed lines show the half max (0.5 probability) for each curve. *p<0.05 for *dmist*[i8/i8] fish compared to wild-type siblings; Kolmogorov–Smirnov test.

The online version of this article includes the following figure supplement(s) for figure 3:

**Figure supplement 1.** CRISPR-generated *dmist* mutants have reduced *dmist* transcript levels.

similarities between Dmist and Fxyd1 (*Jumper et al., 2021*), suggesting that Dmist may belong to a class of small, single-pass transmembrane ion pump regulators.

Using in situ hybridisation, we found that *fxyd1* is expressed in cells along the brain ventricle and choroid plexus (*Figure 4C*) in contrast to the neuronal expression of *dmist* (*Figure 2D*). Despite these different expression patterns, based on their sequence similarity we reasoned that Fxyd1 and Dmist may regulate the same molecular processes that are involved in sleep. To test this hypothesis, we used CRISPR/Cas9 to generate a 28 bp deletion in the third exon of the zebrafish *fxyd1* gene, causing a frameshift that is predicted to encode a truncated protein that lacks the FXYD, transmembrane, and C-terminal domains (*Figure 4B*). Contrary to a previous report based on morpholino knockdown (*Chang et al., 2012*), *fxyd1*[Δ28/Δ28] larvae were viable with no detectable defect in inflation of the brain ventricles. We therefore tested *fxyd1* mutant larvae for sleep phenotypes. Like *dmist* mutants, *fxyd1*[Δ28/Δ28] larvae slept less at night (*Figure 4D–F*). Interestingly, this sleep loss is mainly due to shorter sleep bouts (*Figure 4F*), indicating that *fxyd1* mutants initiate sleep normally but do not properly maintain it, unlike *dmist* mutants, which initiate fewer night-time sleep bouts, although in both cases there is consolidation of the wake state at night (*Figures 3I and 4F*). Thus, despite the non-neuronal expression of *fxyd1* in the brain, mutation of the gene most closely related to *dmist* results in a similar sleep phenotype.

## The brain-wide Na$^+$/K$^+$ pump alpha subunit Atp1a3a regulates sleep at night

Given the similarity between Dmist and Fxyd1 and their effects on night-time sleep, we hypothesised that mutations in Na$^+$/K$^+$ pump subunits known to interact with Fxyd1 might also affect sleep. Consistent with this hypothesis, a low dose of the Na$^+$/K$^+$ pump inhibitor, ouabain, reduced night-time sleep in dose–response studies (*Figure 5—figure supplement 1A*). When applied in the late afternoon of 6 dpf, 1 μM ouabain decreased subsequent night-time sleep by 16.5% relative to controls, an effect size consistent with those observed in *dmist* mutants (*Figure 5A and C*). Night-time waking activity was also significantly increased after low-dose ouabain exposure (*Figure 5B and D*). Ouabain binds to specific sites within the first extracellular domain of Na$^+$/K$^+$ pump alpha subunits (*Price and Lingrel, 1988*), and species-specific changes to these sites confer species-specific ouabain resistance, as in the case of two naturally occurring amino acid substitutions present in the Atp1a1 subunit of mice (*Dostanic et al., 2004*). Alignment of the ouabain sensitive region of zebrafish and mouse Na$^+$/K$^+$ pump alpha subunits revealed that zebrafish Atp1a1a lacks the conserved glutamine at position 121 (*Figure 5E*), suggesting that one of the other subunits with conserved ouabain-binding sites is responsible for the low-dose ouabain sleep effects. We focused on the Na$^+$/K$^+$ pump alpha-3 subunit (Atp1a3) as this has been shown to directly interact with Fxyd1 in mammalian brain tissue (*Feschenko et al., 2003*). Murine *Dmist* expression also correlates well with the *Atp1a3* distribution across five brain cell types in mouse (Pearson correlation coefficient = 0.63), which has the strongest correlation score with neuronal markers (*Figure 5—figure supplement 1B* compared to *Figure 2—figure supplement 1F*). In contrast, zebrafish *atp1a2a* is reportedly expressed in muscle at larval stages, while *atp1a1b* is confined to cells along the ventricle (*Thisse et al., 2001*).

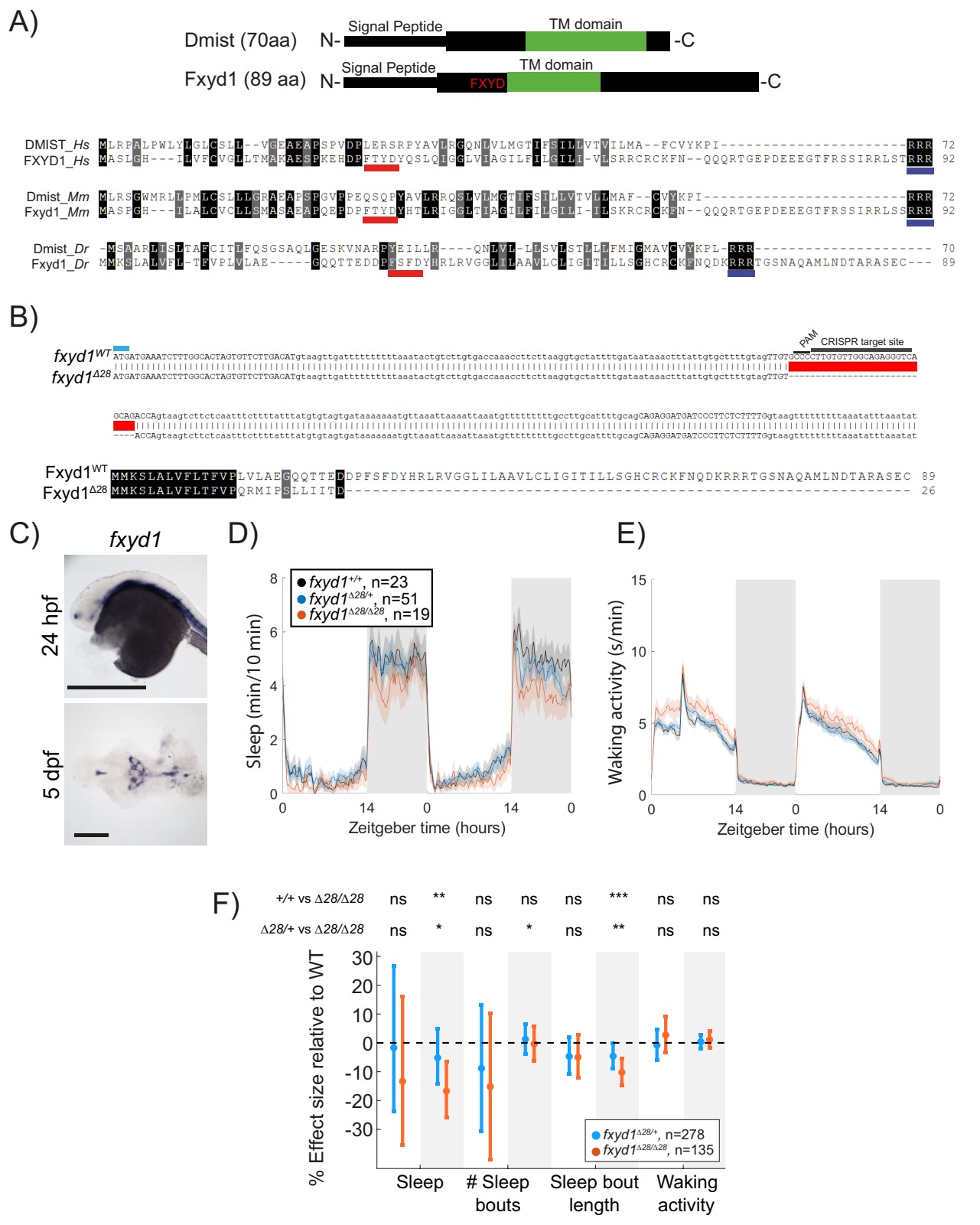

**Figure 4.** Mutation of the *dmist*-related gene *fxyd1* causes reduced sleep at night. (**A**) Schematic of zebrafish Dmist and Fxyd1 protein domains and alignments comparing human, mouse, and zebrafish Dmist and FXYD1 protein sequences. Black and grey shading indicate amino acid identity and similarity, respectively. The FXYD domain is indicated with a red line and the RRR motif in the C-terminus is indicated with a dark blue line. (**B**) CRISPR-Cas9 targeting of the third exon of *fxyd1* created a 28 bp deletion, resulting in a predicted truncated protein. The start codon is marked

*Figure 4 continued on next page*

*Figure 4 continued*

by a cyan line. Guide RNA target sequence and PAM sequence are shown as black bars. The mutant deleted sequence is indicated with a red bar. (**C**) In situ hybridisation of *fxyd1* at 24 hpf (whole animal) and 5 dpf brain (ventral view). Anterior is to the left. Scale bar = 0.5 mm (24 hpf); 0.1 mm (5 dpf). (**D, E**) Representative behavioural experiment showing *fxyd1*$^{\Delta28}$ mutants have decreased night-time sleep (**D**) but normal waking activity at night (**E**). (**F**) Combining five independent experiments with a linear mixed effects model reveals *fxyd1*$^{\Delta28/\Delta28}$ larvae sleep significantly less at night due to shorter sleep bouts compared to *fxyd1*$^{+/+}$ siblings. Plotted are the genotype effect sizes (95% confidence interval) on each parameter relative to wild type. Shading indicates day (white) and night (grey). p-Values are assigned by an *F*-test on the fixed effects coefficients from the linear mixed effects model. *p<0.05, **p<0.01, ***p<0.001, ns p>0.05.

Zebrafish have two Atp1a3 paralogs, *atp1a3a* and *atp1a3b*. Similar to *dmist*, *atp1a3a* is widely expressed in the larval zebrafish brain (*Figure 5F*, compare to *Figure 2D*). While *atp1a3b* is also expressed in the zebrafish brain, its expression is more limited to regions of the midbrain and hindbrain (*Figure 5—figure supplement 1C*). To test whether these genes are involved in regulating zebrafish sleep, we used CRISPR/Cas9 to isolate an allele of *atp1a3a* containing a 19 bp deletion and an allele of *atp1a3b* containing a 14 bp deletion. Both mutations are predicted to generate null alleles due to deletion of the start codon (*Figure 5G*, *Figure 5—figure supplement 1D*). Both *atp1a3a*$^{\Delta19/\Delta19}$ and *atp1a3b*$^{\Delta14/\Delta14}$ mutant larvae were healthy and viable through early development, although *atp1a3b* mutant larvae were not obtained at Mendelian ratios (55 wild type [52.5 expected], 142 [105] *atp1a3b*$^{+/-}$, 13 [52.5] *atp1a3b*$^{-/-}$; p<0.0001, chi-squared), suggesting some impact on early stages of development leading to lethality. Contrary to a previous report based on morpholino injections (*Doğanli et al., 2013*), neither mutant had defects in the inflation of their brain ventricles. Sleep-wake-tracking experiments found that *atp1a3b*$^{\Delta14/\Delta14}$ mutants were more active during the day with minimal sleep phenotypes (*Figure 5—figure supplement 1E–G*). In contrast, mutation of *atp1a3a* resulted in large effects on sleep-wake behaviour. Compared to wild type and heterozygous mutant siblings, *atp1a3a*$^{\Delta19/\Delta19}$ animals were hyperactive throughout the day and night and had a large reduction in sleep at night (*Figure 5H, I*). The night-time sleep reduction was due to a reduction in the length of sleep bouts as *atp1a3a* mutants even had a small increase in the number of sleep bouts at night (*Figure 5J*). In conclusion, loss of *atp1a3a* results in sleep loss at night, similar to treatment with the small molecule *ouabain*, and to *dmist* and *fxyd1* mutants. Notably, the *atp1a3a* mutant phenotype is much stronger, as might be expected if Dmist plays a modulatory, and Atp1a3a a more central, role in Na$^+$/K$^+$ pump activity.

## Dmist modulates Na$^+$/K$^+$ pump function and neuronal activity-induced sleep homeostasis

The similar night-time reduction in sleep in *dmist* and *atp1a3a* mutants, combined with the similarities between Dmist and Fxyd1, suggested that Dmist may regulate the Na$^+$/K$^+$ pump. We therefore exposed wild type and mutant larvae to pentylenetetrazol (PTZ), a GABA-receptor antagonist that leads to globally heightened neuronal activity and elevated intracellular sodium levels that must be renormalised by Na$^+$/K$^+$ pump activity. Consistent with the hypothesis that Dmist and Atp1a3a subunits are important for a fully functional Na$^+$/K$^+$ pump, brains from both *dmist*$^{i8/i8}$ and *atp1a3a*$^{\Delta19/\Delta19}$ larvae had elevated intracellular sodium levels after exposure to PTZ (*Figure 6A*). Thus, neither *dmist* nor *atp1a3a* mutants were able to restore intracellular sodium balance after sustained neuronal activity as quickly as wild-type siblings. Consistent with the night-specific alterations in sleep behaviour, we also found that baseline brain Na$^+$ levels in *dmist* mutants were significantly elevated at night but not during the day (*Figure 6B*). Collectively, these data are consistent with the hypothesis that night-time sleep duration is affected by changes in Na$^+$/K$^+$ pump function and that Dmist is required to maintain this function both at night and after sustained high levels of neuronal activity.

We have previously shown in zebrafish that a brief exposure to hyperactivity-inducing drugs such as the epileptogenic PTZ or wake-promoting caffeine induces a dose-dependent increase in homeostatic rebound sleep following drug washout that is phenotypically and mechanistically similar to rebound sleep following physical sleep deprivation (*Reichert et al., 2019*). Based on their exaggerated intracellular Na$^+$ levels following exposure to PTZ, we predicted that *dmist* mutants would also have increased rebound sleep in response to heightened neuronal activity. Upon wash-on/wash-off of lower dose (5 mM) PTZ, sleep rebound occurs in approximately 50% of wild-type larvae (*Reichert et al., 2019*; *Figure 6C and D*). In contrast, all *dmist*$^{i8/i8}$ larvae showed increased rebound sleep compared

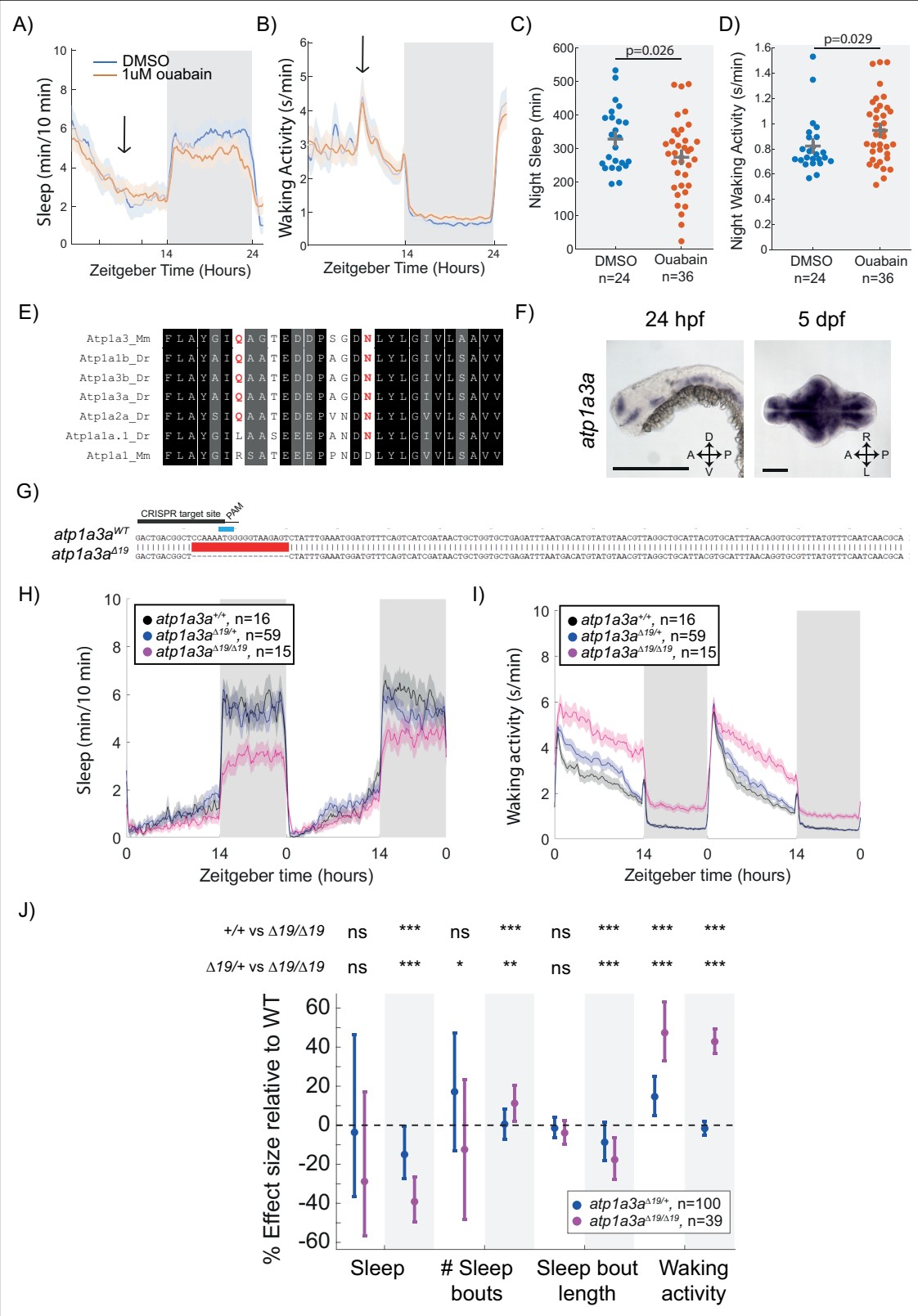

**Figure 5.** Mutation of the Na$^+$/K$^+$ pump alpha subunit *atp1a3a* reduces sleep at night. (**A, B**) Sleep and waking activity traces (± SEM) of wild-type larvae following exposure to 1 μM ouabain. Arrows indicate time the drug was added. (**C, D**) At night, sleep is significantly reduced and waking activity is significantly increased after ouabain exposure. Student's *t*-test, one-tailed. (**E**) Alignments of Na$^+$/K$^+$ pump alpha subunits around the ouabain binding sites. Red indicates residues that are critical for higher sensitivity to ouabain, both of which are present in mouse Atp1a3 but not Atp1a1. (**F**) In situ

*Figure 5 continued on next page*

*Figure 5 continued*

hybridisation of *atp1a3a* at 24 hpf (whole animal) and 5 dpf brain (ventral view). Anterior is to the left. Scale bar = 0.5 mm (24 hpf); 0.1 mm (5 dpf). A, anterior; P, posterior; D, dorsal; V, ventral (**G**) CRISPR-Cas9 targeting of the *atp1a3a* resulted in a 19 bp deletion that eliminates the start codon (blue) and splice junction. Guide RNA target sequence and PAM sequence are shown as black bars. Sequence that is deleted in the mutant is indicated with a red bar. (**H, I**) Representative behavioural experiment showing *atp1a3a*$^{Δ19/Δ19}$ fish are hyperactive throughout the day-night cycle and have decreased sleep at night. Mean ± SEM are shown. (**J**) *atp1a3a*$^{Δ19/Δ19}$ larvae sleep less at night due to shorter sleep bouts. Plotted are the genotype effect sizes (95% confidence interval) on each parameter relative to wild type. Shading indicates day (white) and night (grey). p-Values are assigned by an *F*-test on the fixed effects coefficients from the linear mixed effects model. *p<0.05, **p<0.01, ***p<0.001, ns p>0.05.

The online version of this article includes the following figure supplement(s) for figure 5:

**Figure supplement 1.** Ouabain dose curve and effects of *atp1a3b* mutation on behaviour.

to *dmist*$^{+/+}$ sibling controls (**Figure 6C–E**). Taken together with the elevated sodium retention experiments, such increases in rebound sleep induced by neuronal activity suggests that *dmist*$^{i8/i8}$ fish more rapidly accumulate sleep pressure in response to heightened neuronal activity.

Finally, we predicted that if Dmist is affecting baseline sleep via modulation of Atp1a3a-containing Na$^+$/K$^+$ pumps, *dmist*$^{-/-}$; *atp1a3a*$^{-/-}$ double mutants should have a reduction in night-time sleep that is not the sum of effects from either mutant alone. In other words, if Dmist and Atp1a3a are acting in separate pathways, the double mutant would have an additive phenotype, but if Dmist and Atp1a3a act together in the same complex/pathway, the mutant phenotypes should be non-additive. Indeed, *dmist*$^{-/-}$; *atp1a3a*$^{-/-}$ mutants have a sleep reduction similar to that of *atp1a3a*$^{-/-}$ mutants alone, consistent with a non-additive effect (**Figure 6F**, **Figure 6—figure supplement 1**). Similar non-additivity can be also observed in the *dmist*$^{-/-}$; *atp1a3a*$^{+/-}$ animals, which, like *atp1a3a*$^{+/-}$ animals alone, have a milder sleep reduction, indicating that the lack of additivity between dmist and atp1a3a phenotypes is unlikely due to a floor effect, since double homozygous mutants can sleep even less (**Figure 6F**). This genetic interaction data is consistent with our hypothesis that Atp1a3a and Dmist act in the same pathway—the Na$^+$/K$^+$ pump—to influence sleep.

## Discussion

### Genetic screening discovers *dmist*, a novel sleep-regulatory gene

Using a reverse genetic viral screening strategy, we discovered a short-sleeping mutant, *dmist*, which has a disruption in a previously uncharacterised gene encoding a small transmembrane peptide. Given that the *dmist* mutant appeared within the limited number of 26 lines that we screened, it is likely that many other sleep genes are still waiting to be discovered in future screens. In zebrafish, one promising screening strategy will be to employ CRISPR/Cas9 genome editing to systematically target candidate genes. Advances in the efficiency of this technology now makes it feasible to perform a CRISPR 'F0 screen' in which the consequences of biallelic, gene-specific mutations are rapidly tested in the first generation, with only the most promising lines pursued in germline-transmitted mutant lines (*Grunwald et al., 2019*; *Jao et al., 2013*; *Kroll et al., 2021*; *Shah et al., 2015*; *Shankaran et al., 2017*; *Wu et al., 2018*). CRISPR F0 screens could be scaled to systematically target the large number of candidate sleep-regulatory genes identified through human GWAS studies and sequencing of human patients suffering from insomnia and neuropsychiatric disorders (*Allebrandt et al., 2013*; *Dashti et al., 2019*; *Jansen et al., 2019*; *Jones et al., 2019*; *Lane et al., 2019*; *Lek et al., 2016*; *Palagini et al., 2019*).

### Dmist is related to the Na+/K+ pump regulator Fxyd1

The small Dmist transmembrane protein is highly conserved across vertebrates, expressed in neurons, and important for maintaining normal sleep levels. How can such a small, single-pass transmembrane protein lacking any clear functional domains modulate the function of neurons and ultimately animal behaviour? The recognition that Dmist has sequence homology (~35% amino acid similarity; a conserved 'RRR' motif in the C-terminus) and structural homology (e.g. signal peptide and single-pass transmembrane domains) to the Na$^+$,K$^+$-ATPase pump regulator Fxyd1 (Phospholemman) offers some important clues.

Fxyd1/Phospholemman is a member of the FXYD protein family, of which there are seven mammalian members (*Sweadner and Rael, 2000*). Each of the FXYD proteins is small, contains a characteristic

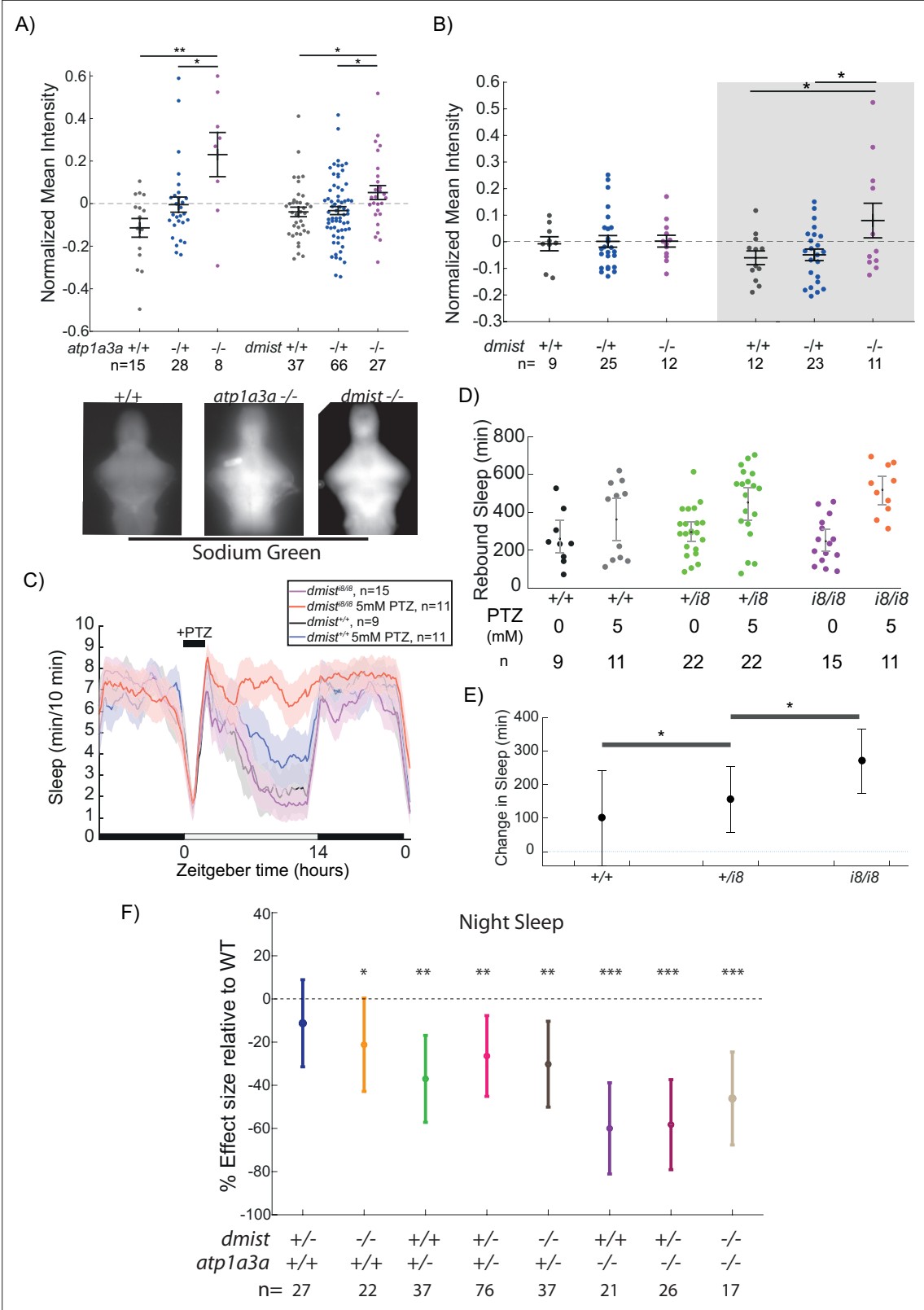

**Figure 6.** *dmist* mutants have altered sodium homeostasis. (**A**) Brain sodium levels are significantly elevated after exposure to pentylenetetrazol (PTZ) in both *atp1a3a^{Δ19/Δ19}* (two independent experiments) and *dmist^{i8/i8}* (four independent experiments) fish relative to wild type and heterozygous mutant siblings, as measured by fluorescence intensity of Sodium Green, normalised to the sample mean intensity. Crosses show mean ± SEM. n indicates the number of animals. Below are example images of brains stained with Sodium Green. *p<0.05, **p<0.01, one-way ANOVA, Tukey's post

*Figure 6 continued on next page*

*Figure 6 continued*

hoc test. (**B**) Under baseline conditions, brain sodium levels are significantly elevated in *dmist*$^{i8/i8}$ fish at night but not during the day, as measured by fluorescence intensity with Sodium Green. Crosses show mean ± SEM. *p<0.05, **p<0.01, one-way ANOVA, Tukey's post hoc test. (**C**) *dmist*$^{i8/i8}$ larvae have increased rebound sleep compared to wild-type siblings following exposure to 5 mM PTZ. Representative sleep traces of *dmist*$^{+/+}$ (no drug, water vehicle controls in black; PTZ exposed in blue) and *dmist*$^{i8/i8}$ (no drug in purple; PTZ exposed in red) following 1 hr exposure to 5 mM PTZ (black bar) in the morning. Data are mean ± SEM. *dmist*$^{i8/+}$ animals are not plotted for clarity but are included in panel (**D**). (**D**) Rebound sleep after exposure to 5 mM PTZ, calculated from the experiment in (**C**). Each dot represents a single fish, grey lines show mean ± SEM. (**E**) Effect size of change in sleep after 1 hr treatment with 5 mM PTZ (and washout) compared to vehicle-treated controls (error bars show 95% confidence intervals). *p<0.05, one-way ANOVA, Tukey's post hoc test. (**F**) Effect sizes (and 95% confidence interval) relative to wild types (dotted line) on sleep at night in larvae from *dmist*$^{+/-}$; *atp1a3a*$^{+/-}$ in-crosses from three independent experiments. p-Values are assigned by an *F*-test on the fixed effects coefficients from the linear mixed effects model relative to *dmist*$^{+/+}$; *atp1a3a*$^{+/+}$ animals. For all sleep-wake parameters, see ***Figure 6—figure supplement 1***. *p<0.05, **p<0.01, ***p<0.0001, ns p>0.05.

The online version of this article includes the following figure supplement(s) for figure 6:

**Figure supplement 1.** Sleep effects in *dmist*$^{-/-}$; *atp1a3a*$^{-/-}$ double mutants are non-additive.

FXYD domain, and has a single transmembrane domain. FXYD family members interact with alpha subunits of the Na$^+$,K$^+$ ATPase to regulate the function of this pump, with individual family members expressed in different tissues to modulate Na$^+$,K$^+$-ATPase activity depending on the physiological needs of the tissue (***Geering et al., 2003***). In cardiac muscle, FXYD1 is thought to act as a hub through which various signalling cascades, such as PKA, PKC, or nitric oxide, can activate or inhibit Na$^+$ pump activity (***Pavlovic et al., 2013***). For example, FXYD1 is critical for mediating the increased Na$^+$ pump activity observed after β-receptor stimulation via cAMP-PKA signalling (***Despa et al., 2008***). Much less is known about the role of FXYD1 in non-cardiac tissue, although it is expressed in neurons in the mammalian cerebellum, the choroid plexus, and ependymal cells, where it interacts with all three alpha subunits of the Na$^+$,K$^+$ ATPase (***Feschenko et al., 2003***).

In zebrafish, we also found that *fxyd1* is expressed in cells around the ventricles and in the choroid plexus (***Figure 4C***), in contrast to *dmist* which is expressed in neurons throughout the brain. Despite the different expression patterns, mutation of each gene resulted in a similar reduction of sleep at night. However, unlike *dmist* mutants, which have fewer sleep bouts (i.e. initiate sleep less) and an increase in waking locomotor activity, *fxyd1* mutants have shorter sleep bouts (i.e. cannot maintain sleep) on average and do not have a locomotor activity phenotype. Just as the various FXYD family members modulate the Na$^+$/K$^+$ pump in different tissue- and context-specific ways, this phenotypic variation between *fxyd1* and *dmist* mutants could be due to the different *fxyd1* and *dmist* expression patterns, modulation kinetics of pump/channel dynamics, or interaction with different accessory proteins or signal transduction cascades. Nevertheless, the similar timing and magnitude of sleep reduction, combined with the structural similarity of Fxyd1 and Dmist, suggest that they may regulate similar sleep-related processes.

## Dmist, the sodium pump, and sleep

The similarity between Dmist and FXYD1 led us to directly manipulate the Na$^+$,K$^+$ ATPase to test its importance in sleep. The Na$^+$,K$^+$-ATPase is the major regulator of intracellular Na$^+$ in all cells and, by actively exchanging two imported K$^+$ ions for three exported Na$^+$ ions, is essential for determining cellular resting membrane potential (reviewed in ***Clausen et al., 2017***). The Na$^+$,K$^+$-ATPase consists of a catalytic alpha subunit (four known isoforms, ATP1A1-4), a supporting beta subunit (three isoforms, ATP1B1-3), and a regulatory gamma subunit (the FXYD proteins). The alpha1 and alpha3 subunits are the predominant catalytic subunits in neurons (alpha2 is mostly restricted to glia), although the alpha1 subunit is also used ubiquitously in all tissues (***McGrail et al., 1991***). By mutating zebrafish orthologs of *Atp1a3*, we therefore could test the neuronal-specific role of the Na$^+$,K$^+$-ATPase in sleep.

Mutations in both zebrafish *Atp1a3* orthologs increased waking locomotor behaviour during the day. However, only mutations in *atp1a3a*, which is expressed brain-wide, but not in *atp1a3b*, which is expressed in more restricted brain regions, led to changes in night-time sleep. The *atp1a3a* mutants have a larger sleep reduction than *dmist*$^{vir}$, *dmist*$^{i8}$, or *fxyd1*$^{\Delta28}$ mutants, which is expected since loss of a pump subunit should have a larger effect than the loss of a modulatory subunit, as has been shown for other ion channels (***Cirelli et al., 2005***; ***Wu et al., 2014***). Autosomal-dominant missense mutations leading to loss of function in *ATP1A3* cause movement disorders such as rapid-onset dystonia parkinsonism and childhood alternating hemiplegia (recurrent paralysis on one side) in humans (***de Carvalho***

*Aguiar et al., 2004*; *Heinzen et al., 2012*; *Heinzen et al., 2014*), while loss-of-function mutations in *Atp1a3* result in generalised seizures and locomotor abnormalities, including hyperactivity, in mice, which was not observed in zebrafish (*Clapcote et al., 2009*; *Hunanyan et al., 2015*; *Ikeda et al., 2013*; *Kirshenbaum et al., 2011*; *Sugimoto et al., 2014*). A very high prevalence of insomnia was recently reported in patients with childhood alternating hemiplegia, some of which harboured mutations in *Atp1a3* (*Kansagra et al., 2019*), consistent with our observations that insomnia at night is a direct behavioural consequence of *atp1a3a* mutation in zebrafish. Since zebrafish *atp1a3a* mutants phenocopy the insomnia and hyperactivity phenotypes observed in patients, small-molecule screens aimed at ameliorating zebrafish *atp1a3a* mutant phenotypes may be a promising approach for the rapid identification of new therapies for the management of this disease (*Hoffman et al., 2016*; *Rihel et al., 2010*).

Together, the night-specific sleep phenotypes of *dmist*, *fxyd1*, and *atp1a3a* mutants point to a role for the Na$^+$,K$^+$-ATPase in boosting sleep at night. How might the alpha3 catalytic subunit of the Na$^+$/K$^+$ pump regulate sleep, and how could Dmist be involved? We found that Dmist is required for proper maintenance of brain intracellular Na$^+$ levels at night but not during the day, mirroring the timing of sleep disruption in *dmist$^{i8/i8}$* animals. This suggests that the decreased night-time sleep of *dmist* mutants is due to a specific requirement for Dmist modulation of the Na$^+$/K$^+$ pump at night. However, we cannot exclude the possibility that Dmist's function is required in only a subset of critical sleep/wake regulatory neurons during the day that then influence behaviour at night, such as the wake-active, sleep-homeostatic regulating serotonergic neurons of the raphe (*Oikonomou et al., 2019*), or wake-promoting Hcrt/orexin neurons (*Li et al., 2022*). We also cannot exclude a role for Dmist and the Na$^+$/K$^+$ pump in developmental events that impact sleep, although our observation that ouabain treatment, which inhibits the pump acutely after early development is complete, also impacts sleep, argues against a developmental role. Another possibility is that disruption of proper establishment of the Na$^+$ electrochemical gradient in *dmist* mutant neurons leads to dysfunction of various neurotransmitter reuptake transporters, including those for glycine, GABA, glutamate, serotonin, dopamine, and norepinephrine, which rely on energy from the Na$^+$ gradient to function (*Kristensen et al., 2011*).

A third possibility is that Dmist and the Na$^+$,K$^+$-ATPase regulate sleep not by modulation of neuronal activity per se but rather via modulation of extracellular ion concentrations. Recent work has demonstrated that interstitial ions fluctuate across the sleep/wake cycle in mice. For example, extracellular K$^+$ is high during wakefulness, and cerebrospinal fluid containing the ion concentrations found during wakefulness directly applied to the brain can locally shift neuronal activity into wake-like states (*Ding et al., 2016*). Given that the Na$^+$,K$^+$-ATPase actively exchanges Na$^+$ ions for K$^+$, the high intracellular Na$^+$ levels we observe in *atp1a3a* and *dmist* mutants are likely accompanied by high extracellular K$^+$. Although we can only speculate at this time, a model in which extracellular ions that accumulate during wakefulness and then directly signal onto sleep-regulatory neurons could provide a direct link between Na$^+$,K$^+$ ATPase activity, neuronal firing, and sleep homeostasis. Such a model could also explain why disruption of *fxyd1* in non-neuronal cells also leads to a reduction in night-time sleep.

In addition to decreased night-time sleep, we also observed that *dmist* mutants have an exaggerated sleep rebound response following the high, widespread neuronal activity induced by the GABA-receptor antagonist, PTZ. Since both Atp1a3a and Dmist were essential for re-establishing proper brain intracellular Na$^+$ levels following PTZ exposure (*Figure 6A*), we speculate that the exaggerated sleep rebound is a consequence of increased neuronal depolarisation due to defective Na$^+$ pump activity. This is consistent with our previous observations that the intensity of brain-wide neuronal activity impacts the magnitude of subsequent sleep rebound via engagement of the Galanin sleep-homeostatic output arm (*Reichert et al., 2019*). Why does loss of *dmist* lead to both decreased night-time sleep and increased sleep rebound in response to exaggerated neuronal activity during the day? One possibility is that Na$^+$/K$^+$ pump complexes made up of different alpha and beta subunits may be differentially required for maintaining Na$^+$ homeostasis under physiological conditions and have different affinities for (or regulation by) Dmist. For example, the Atp1a1 subunit is considered the Na$^+$/K$^+$ pump workhorse in neurons, while Atp1a3, which has a lower affinity for Na$^+$ ions, plays an essential role in repolarising neurons when Na$^+$ rapidly increases during high levels of neuronal activity, such as after a seizure (*Azarias et al., 2013*). If Dmist preferentially interacts with Atp1a3a subunit, with which the non-additive effect of *dmist* and *atp1a3a* mutation on sleep is consistent, day-time sleep-related phenotypes in *dmist* mutants might be uncovered only during physiological

challenge. Conversely, neurons may be more dependent on Atp1a3a and Dmist for sodium homeostasis at night due to changes in Na$^+$/K$^+$ pump composition, Dmist interactions, or ion-binding affinities. For example, activity of the Na$^+$/K$^+$ pump can be modulated by the circadian clock (*Damulewicz et al., 2013*; *Nakashima et al., 2018*), changes in substrate availability, including ATP (reviewed in *Therien and Blostein, 2000*), or hormones (*Ewart and Klip, 1995*). Teasing out how Dmist modulation of the Na+/K+ pump changes across the day-night cycle, and in which neurons Dmist's function may be particularly important at night, will require future investigation.

In conclusion, through a genetic screening strategy in zebrafish, we have identified a novel brain expressed gene that encodes a small transmembrane protein regulator of night-time sleep and wake behaviours. Future work will be required to uncover the precise signalling dynamics by which Dmist regulates the Na$^+$,K$^+$-ATPase and sleep.

# Materials and methods

## Zebrafish husbandry

All zebrafish lines were housed on a 14 hr:10 hr light:dark schedule in dechlorinated water at 27.5°C and routine husbandry was performed by the UCL Zebrafish Facility. Embryos were collected from spontaneous spawning and staged according to *Kimmel et al., 1995*.

Embryos and larvae were raised on a 14 hr:10 hr light:dark schedule in 10 cm Petri dishes at a density of 50 embryos per 10 cm Petri dish. Embryo water (~pH 7.3, temperature 28.5°C, conductivity ~423.7 uS with methylene blue) was changed daily and animals over 4 dpf were euthanised by overdose of MS-222 (300 mg/l) or 15% 2-phenoyethanol (77699, Sigma-Aldrich) at the end of experiments.

Raising of genetically altered zebrafish and all experimental procedures were performed under project licence 70/7612 and PA8D4D0E5 awarded to JR under the UK Animals (Scientific Procedures) Act 1986 guidelines.

## Lines

The *dmist*[vir] allele was generated in wild-type line T/AB-5 (*Varshney et al., 2013*) and outcrossed to Harvard AB. The *dmist*[i8], *fxyd1*[Δ28], *atp1a3a*[Δ19], and *atp1a3b*[Δ14] alleles were generated and maintained at UCL on an AB/TL background (*Table 1*). Both *dmist*[i8] and *dmist*[vir] were outcrossed to the AB strain at UCL for at least three generations.

## Larval zebrafish behavioural tracking

At 4 dpf, zebrafish larvae were placed into individual wells of a 96-square-well plate (WHA7701-1651, Sigma-Aldrich) filled with 650 µl of embryo water per well and tracked for 3 d under a 14hr:10hr light:dark schedule (lights on, 09:00; lights off, 23:00) using automated videotracking in ViewPoint ZebraBoxes (Viewpoint Life Sciences). The 96-well plate was under constant illumination with infrared LEDs, and white LEDs simulated the light:dark schedule. Videography (with one-third-inch Dragonfly2 PointGrey monochrome camera, frame rate: 25–30 Hz; fixed-angle megapixel lens, Computar M5018-MP) of individual behaviour was recorded in quantisation mode to detect movement by background subtraction between frames in individual wells with 60 s integration time bins. The parameters

**Table 1.** Zebrafish lines used in the article.

| Strain designation | Allele number | Gene identifier | Additional information |
|---|---|---|---|
| *10543/dmist*[vir] | *la015577Tg* | ENSDARG00000095754 | Maintained at UCL |
| *dmist*[i8] | *u505* | ENSDARG00000095754 | Maintained at UCL |
| *fxyd1*[Δ28] | *u504* | ENSDARG00000099014 | Maintained at UCL |
| *atp1a3a*[Δ19] | *u513* | ENSDARG00000018259 | Maintained at UCL |
| *atp1a3b*[Δ14] | *u514* | ENSDARG00000104139 | Maintained at UCL |

used for detection were calibrated according to the sensitivity of individual boxes but were in the following range: detection threshold, 15–20; burst, 50 pixels; and freeze, 3–4 pixels. Embryo water in the wells was topped up daily with fresh water, and ambient room temperature was maintained at approximately 26°C. Output data was sorted, parsed, and analysed by custom Perl and MATLAB scripts (MATLAB R2016 version 9.1, The MathWorks), as in *Rihel et al., 2010*.

Oxygen-permeable lids (Applied Biosystems, 4311971) were applied over the top of the 96-well plate when performing experiments in constant darkness, and the larvae were left undisturbed for the duration of the experiment to avoid light exposure.

At the end of the experiment, all larvae were visually checked for health before euthanasia and transfer to individual wells of a 96-well PCR plate for DNA extraction and genotyping.

## Behavioural analysis

Sleep parameters were calculated as in *Rihel et al., 2010*. For each genotype, exemplar experiments are shown, and summary data was analysed by combining experiments with an LME model as follows. Behavioural summaries across multiple experiments were determined using the MATLAB fitlme function to fit an LME model for each parameter with genotype as a fixed effect and independent experiment as a random effect, then representing the effect size as a % change from the wild-type value. Before fitting the LME model, the parameters sleep, sleep length, and waking activity were log normalised by calculating the log of 1+ the parameter value for each larva.

Circadian period for every larva was calculated using the MATLAB findpeaks function on the activity (delta-pixels) time-series data with a minimum peak distance of 18 hr (1080 min). N-way ANOVA was calculated to evaluate differences between groups.

## Adult behavioural tracking

Fish from a *dmist^{i8/+}* × *dmist^{i8/+}* cross were raised in a mixed gender tank to adulthood. Zebrafish adults (aged 3–4 mo) were randomly selected and tracked on a 14hr:10hr light:dark cycle (180 lux at water surface, lit from above) for 3 d as in *Chiu et al., 2016*. In brief, fish were placed into uncovered plastic chambers (7 × 12 × 8.5 cm; W × L × H) with small holes for water exchange, and these were placed in a circulating water tank (46 × 54 cm with 4.5 cm water height). This setup was supplied with fish water from the home aquarium heated to 28°C and pumped from a 45 l reservoir at a flow rate of 1.3 l/min. Infrared light (60°, 54 LED Video Camera Red Infrared Illuminator Lamp, SourcingMap, with the ambient light detector covered) was continuously supplied from below. Fish were tracked at 15 Hz using Viewpoint Life Sciences ZebraBox tracking software in tracking mode, with a background threshold of 40, inactive cutoff of 1.3 cm/s, and a small movement cutoff of 8 cm/s. Each track was visually inspected for errors at 1 min resolution across the entire session and analysed using custom MATLAB scripts (MATLAB R2016 version 9.1, The MathWorks, Inc). Experiments were performed blind to genotype, which was determined by fin-clip after the experiment. Females and males were originally analysed separately; since no significant gender effect was found (two-way ANOVA, genotype × gender), data from both genders were pooled for the final analysis.

## Genotyping

Prior to genotyping, adult fish were anaesthetised in 30 µg/ml MS-222, fin-clipped by cutting a small section of the caudal fin, and then allowed to recover in fresh fish water. For pooled experiments, 3 dpf larvae from heterozygous in-crosses were fin-clipped as in *Wilkinson et al., 2013* and allowed to recover in a square 96-well plate to keep larvae separate prior to pooling larvae of the same genotype. Genomic DNA was extracted from adult fin clips and larvae by boiling for 30 min in 50 µl 1× base solution (0.025 M KOH, 0.2 mM EDTA). Once cooled, an equal volume (50 µl) of neutralisation buffer (0.04 M Tris-HCl) was then added and undiluted genomic DNA used for genotyping.

The *dmist^{vir}* genotype was detected by PCR (standard conditions) using a cocktail of three primers (0.36 mM final concentration of each primer) to detect the wild-type allele and viral insertion (see *Table 2*) so that genotypes could be assigned according to size of bands detected (*dmist^{vir/vir}* 800 bp; *dmist^{vir/+}* 508 bp and 800 bp; *dmist^{+/+}* 508 bp).

The *dmist^{i8}* genotype was assigned by KASP genotyping using allele-specific primers (*dmist^{i8}* allele 5′-GATCTCCCT[GCAGAAAGAT]CTTTCTGCA-3′=FAM, *dmist^+* allele 5+-GATCTCCCT[CACCG]

**Table 2.** Primer sequences used in the article.

| | Oligo name | Sequence (5' -> 3') | Anneal temperature (°C) | Application |
|---|---|---|---|---|
| 1 | dmist_vir_fw | CACAGGGATGTGTGATGCCGGTTAAC | 55 | dmistvir genotyping |
| 2 | dmist_vir_rev | GTAGACACATACTGCCATACCAATC | 55 | dmistvir genotyping |
| 3 | vir_fw | CACCAGCTGAAGCCTATAGAGTACGAGC- | 55 | dmistvir genotyping |
| 4 | dmist_Dr_5RACE_fw | CGTTTCGCCACAATGTCAGCA | 55–65 | dmist_Dr 5'RACE |
| 5 | dmist_Dr_5RACE_rev_outer | AATGTTCAACTCCAGGCGTC | 55–65 | dmist_Dr 5'RACE |
| 6 | dmist_Dr_5RACE_rev_inner | AATGTTCAACTCCAGGCGTC | 55–65 | dmist_Dr 5'RACE |
| 7 | dmist_Dr_3RACE_fw_inner | GACGCCTGGAGTTGAACATT | 55–65 | dmist_Dr 3'RACE |
| 8 | dmist_Dr_3RACE_fw_outer | GGTATGGCAGTATGTGTCTACA | 55–65 | dmist_Dr 3'RACE |
| 9 | Dmist_Mm_3RACE_outer | GCTGGTGACTGTCCTCCTTATG | 55–65 | dmist_Mm 3'RACE |
| 10 | Dmist_Mm_3RACE_inner | GTGTCTACAAGCCCATCCGTC | 55–65 | dmist_Mm 3'RACE |
| 11 | dmist_Dr_fw | TTTCGCCACAATGTCAGCAGC | 56 | dmist_Dr probe |
| 12 | dmist_Dr_rev | CGACTTTCATTTATTAGTTCAGACATGTC | 56 | dmist_Dr probe |
| 13 | qPCR_dmist_fw | ACGCCAGACCTTATGAAATCC | 60 | RT-qPCR |
| 14 | qPCR_dmist_rev | TGCGTCGGAGAGGGTTTGTAG | 60 | RT-qPCR |
| 15 | qPCR_ankrd13a_fw | TGGTGGCGTTCCAGAGTTAC | 60 | RT-qPCR |
| 16 | qPCR_ankrd13a_rev | GGACACGAGAGGAATCCAGC | 60 | RT-qPCR |
| 17 | qPCR_slc6a4b_fw | ACATGGTTGGGTCGACGTTT | 60 | RT-qPCR |
| 18 | qPCR_slc6a4b_rev | TCCAACCCACCAAAAGTGCT | 60 | RT-qPCR |
| 19 | ef1alpha_fw | TGCTGTGCGTGACATGAGGCAG | 60 | RT-qPCR |
| 20 | ef1alpha_rev | CCGCAACCTTTGGAACGGTGT | 60 | RT-qPCR |
| 21 | SP6dmist_sgRNA | ATTTAGGTGACACTATAGCGTTATGCAGAAAGCGGTGGTTTTAGAGCTAGAAATAGCAAG | n/a | CRISPR |
| 22 | T7atp1a3a_sgRNA | TAATACGACTCACTATAGACTGACGGCTCCAAAATGGGTTTTAGAGCTAGAAATAGCAAG | n/a | CRISPR |
| 23 | SP6fxyd1_sgRNA | ATTTAGGTGACACTATAGGACCCCTGCCAACACAAGGTTTTAGAGCTAGAAATAGCAAG | n/a | CRISPR |
| 24 | SP6atp1a3b_sgRNA | ATTTAGGTGACACTATAGGACTGACGCGCAACCATGGTTTTAGAGCTAGAAATAGCAAG | n/a | CRISPR |
| 25 | HRM_dmist_fw | GCCACAATGTCAGCAGCACG | 59 | HRM |
| 26 | HRM_dmist_rev | GCGTTCACTTTAGACTCTCCCAGC | 59 | HRM |
| 27 | HRM_atp1a3a_fw | TGACAGACTGAAGAAACAGC | 55 | HRM |
| 28 | HRM_atp1a3a_rev | TTAAATCTCAGCACCACCAGCAG5 | 55 | HRM |
| 29 | HRM_fxyd1_fw | TGACCAAACCTTCTTAAGGTGC | 58 | HRM |

*Table 2 continued on next page*

*Table 2 continued*

| | Oligo name | Sequence (5' -> 3') | Anneal temperature (°C) | Application |
|---|---|---|---|---|
| 30 | HRM_fxyd1_rev | AAATTGAGAAGACTTACTGGTCTGC | 58 | HRM |
| 31 | HRM_atp1a3b_fw | AAAGGCTGTCACTTTCTCCATCAC5 | 58 | HRM |
| 32 | HRM_atp1a3b_rev | TGCAGTAGATGAGGAATCGGTC | 58 | HRM |
| 33 | MiSeq_dmist_fw | TCGTCGGCAGCGTCAGATGTGTATAAGAGACAGTATAACTTACGTGTGGACGGACTC | 58 | MiSeq |
| 34 | MiSeq_dmist_rev | GTCTCGTGGGCTCGGAGATGTGTATAAGAGACAGTTGCCTCAGCAGGATTTCATAAG | 58 | MiSeq |
| 35 | MiSeq_atp1a3a_fw | TCGTCGGCAGCGTCAGATGTGTATAAGAGACAGTCGTTATCCGTGCAAGAGCTTC | 58 | MiSeq |
| 36 | MiSeq_atp1a3a_rev | GTCTCGTGGGCTCGGAGATGTGTATAAGAGACAGTTCTCAGCACCAGCAGTTATCG | 58 | MiSeq |
| 37 | | TCGTCGGCAGCGTCAGATGTGTATAAGAGACAGTGACTGACATTCTCTTCGTG | 58 | MiSeq |
| 38 | MiSeq_atp1a3b_fw MiSeq_atp1a3b_rev | GTCTCGTGGGCTCGGAGATGTGTATAAGAGACAGTTCTCGTGATGCAGTAGATGAGG | 68 | MiSeq |
| 39 | MiSeq_fxyd1_fw | TCGTCGGCAGCGTCAGATGTGTATAAGAGACAGAAAATACTGTCTTGTGACCAAACC | 57 | MiSeq |
| 40 | MiSeq_fxyd1_rev | GTCTCGTGGGCTCGGAGATGTGTATAAGAGACAGTTCATCCTGCTGCAAAATGC | 57 | MiSeq |
| 41 | attB1-dreammist forward primer | GGGGACAAGTTTGTACAAAAAAGCAGGCTTCACCATGTCAGCAGCACGCCTGATCTCC | 55–60 | Gateway |
| 42 | attB3-dreammist reverse primer | GGGGACCACTTTGTACAAGAAAGCTGGGTATCACCTGCGTCGGAGAGGTTTGTAG | 55–60 | Gateway |
| 43 | Dmist-GFPA22WFw | GCTTTTCCAGTCTGGGAGTT**GG**CAGCTGGGAGAGTCTAAAG | 66 | SDM |
| 44 | Dmist-GFPA22WRev | CTTTAGACTCTCCCAGCTG**CCA**ACTCCCAGACTGGAAAAGC | 66 | SDM |

CTTTCTGCA-3′=HEX; KASP master mix KBS-1016-011) and assay were prepared and analysed according to the manufacturer's protocol (LGC Genomics).

The *atp1a3a$^{Δ19}$* genotype was assigned by KASP genotyping using allele-specific primers (*atp1a3a$^{Δ19}$* allele 5′-GACAGACTGAAGAAACAGCGACTGACGGCTC[CAAAATGGGGGTAAGAGTC]–3′=FAM, *atp1a3a$^+$* allele 5′-GACAGACTGAAGAAACAGCGACTGACGGCTC-3[]=HEX).

The *atp1a3b$^{Δ14}$* genotype was assigned by PCR using MiSeq_atp1a3b primers (*Table 2*), with the *atp1a3b$^{Δ14}$* allele running 14 bp faster than the *atp1a3b$^+$* allele.

*fxyd1$^{Δ28}$* was assigned by KASP genotyping using allele-specific primers (*fxyd1$^{Δ28}$* allele 5′-GAAGGTCGGAGTCAACGGATTTAATAAACTTTATTGTGCTTTTGTAGTTGT[A]–3′=HEX, *fxyd1$^+$* allele 5+-GAAGGTGACCAAGTTCATGCTTAATAAACTTTATTGTGCTTTTGTAGTTGT[G]–3′=FAM) or PCR using MiSeq_fxyd1 primers (see *Table 2*) followed by digestion with the restriction enzyme DrdI, which yields bands at 138 bp and 133 bp for *fxyd1$^{+/+}$*; 138 bp, 133 bp and 271 bp for *fxyd1$^{+/Δ28}$*, and 243 bp for *fxyd1$^{Δ28}$*.

## 3′ RACE

FirstChoice RLM-RACE kit (Ambion AM1700) was used to amplify the 5′ and 3′ ends from cDNA obtained from 4 dpf larvae raised on a 14:10 LD cycle and C57BL/6 E13.5 mouse embryos obtained from the Parnavalas lab (UCL). 5′ and 3′ RACE primers were designed according to the manufacturer's guidelines (*Table 2*) and the manufacturer's protocol was followed. Clones were sequenced by Sanger sequencing.

## In situ hybridisation

Probes were designed to target the 3′UTR and entire ORF of *dmist_Dr* transcript using primers that amplified the target region from zebrafish cDNA under standard PCR conditions (expected size 1325 bp; *Table 2*). The PCR product was cloned into pSC vector (Strataclone PCR cloning kit Agilent 240205-12) and verified by Sanger sequencing. Antisense probe was generated by cleavage of pSC-dmist plasmid with XbaI and in vitro transcribed with T3 polymerase (Promega P2083) using 1 µg DNA template according to the standard in vitro transcription protocol (see the full protocol at https://dx. doi.org/10.17504/protocols.io.ba4pigvn). RNA probe was extracted and purified using the ZYMO RNA concentrator kit (Zymo #R1013).

Whole-mount in situ hybridisation was performed according to *Thisse and Thisse, 2008* with the following adaptations. Embryos <5 dpf were dechorionated and fixed at the appropriate stage in 4% paraformaldehyde (PFA) overnight at 4°C. 5 dpf larvae were fixed in 4% PFA/4% sucrose overnight at 4°C and then washed 3 × 5 min in PBS prior to dissecting out the brain. Fixed embryos were washed 3 × 5 min in PBS, progressively dehydrated into 100% methanol (MeOH), and stored at –20°C overnight. Prior to pre-hybridisation, embryos were bleached for 30 min in the dark (0.05% formamide, 0.5× SSC, 6% $H_2O_2$) and then fixed in 4% PFA for 30 min at room temperature. To image, the embryos were progressively rehydrated into 0.1% PBTw, progressively sunk in to 80% glycerol, and imaged on a Nikon compound microscope (Nikon Eclipse Ni, Leica MC190HD camera).

## RT-qPCR

Larvae from heterozygous in-crosses (*dmist$^{i8/+}$* or *dmist$^{vir/+}$*) were genotyped by tail biopsy at 3 dpf (*Wilkinson et al., 2013*) and allowed to recover fully in individual wells of a square welled 96-well plate before euthanising at 5 dpf. RNA was extracted from three 5 dpf embryos of each genotype by snap freezing in liquid nitrogen and TRIzol RNA extraction (Ambion 15596026) with the following modifications to the manufacturer's protocol: 400 µl total TRIzol reagent used to homogenise larvae using a pellet pestle homogeniser, and 5 µg glycogen (Invitrogen, Cat# 10814010; 20 µg/µl) was added to the RNA solution after chloroform extraction to aid precipitation of the RNA. The cDNA library was synthesised from high-quality RNA (Agilent AffinityScript qPCR cDNA synthesis kit 600559), diluted 1:10, and gene-specific primers (*Table 2*) were used for amplification of target genes with SYBR green mastermix in a Bio-Rad CFX Real-Time qPCR instrument. Gene expression levels were normalised to the housekeeping gene *ef1alpha* (primers in *Table 2*) and analysed using custom MATLAB scripts (MATLAB v9.2 2017, The MathWorks 2017).

## Sodium Green assay

Cell permeable Sodium Green tetraacetate (Invitrogen, S6901) was prepared fresh from frozen stock by dissolving in DMSO to 1 mM then diluting in fish water to a final concentration of 10 µM. About 50 larvae (5–7 dpf) from $atp1a3a^{\Delta19/+}$ or $dmist^{i8/+}$ in-crosses were placed in wells of a 6-well plate, then most fish water was removed and replaced with 3 ml of the 10 µM Sodium Green solution for 2 hr. During exposure, the plate was covered in foil and placed in a 28°C incubator. For PTZ experiments, larvae were also exposed to 10 mM PTZ (diluted from 1 mM stock dissolved in water) for 2 hr. For timepoints at night (ZT17-19), larvae were handled and collected under red light. After soaking in Sodium Green, larvae were washed 3× with fish water, anaesthetised with MS-2222, and fixed in 4% PFA/4% sucrose overnight at 4°C. After 3× wash in PBS, larval brains were dissected and placed in 200 µl PBS in a 48-well plate, and the matched bodies were used for genotyping (see 'Genotyping'). Brains were imaged using an upright MVX10 MacroView microscope with an MC PLAPO ×1 objective (both OLYMPUS) with a mercury lamp for fluorescent excitation at 488 nm (OLYMPUS, U-HGLGPS). Images of roughly the same focal plane (dorsal/ventral view) were taken with an XM10 OLYMPUS camera by a single exposure following minimal light exposure (to avoid bleaching). Mean fluorescent intensity was calculated from ROIs placed on the optic tectum/midbrain using ImageJ, background subtracted, and normalised to the average fluorescence intensity for each imaging session.

## Protein alignments

Cross-species *dmist* homologues were identified by reciprocal BLASTp of the C-terminal region of Dmist_*Dr* in vertebrate genomes. Translations of candidate transcript ORFs were then aligned with Dmist_*Dr* using ClustalOmega to calculate the percentage identity matrix (https://www.ebi.ac.uk/Tools/msa/clustalo/) and visualised with the tool Multiple Align Show (https://www.bioinformatics.org/sms/multi_align.html).

To identify Dmist orthologs, Dmist peptides were aligned with the multiple sequence alignment tool MAFFT (*Katoh and Toh, 2010*) and seeded into a JackHMMR iterative search of the UniProt database (*Johnson et al., 2010*). Protein-protein alignments of Dmist to Fxyd1 were then performed using ClustalOmega and visualised with the tool Multiple Align Show.

## CRISPR/Cas9 gene targeting

CRISPR targets were designed and synthesised according to *Gagnon et al., 2014* using ChopChop (*Montague et al., 2014*; http://chopchop.cbu.uib.no/; see *Table 2* for sequences) to identify target sites. 100 pg sgRNA and 300 pg Cas9 mRNA (pT3TS-nCas9n) were injected into the yolk of one-cell stage AB-TL embryos obtained from natural spawning. F0 fish were screened by high-resolution melt (HRM) analysis using gene-specific primers (*Table 2*) with Precision melt supermix (Bio-Rad 1725112) according to the manufacturer's protocol in a Bio-Rad CFX RT-PCR thermocycler. Positive founders identified in HRM analysis were then sequenced by Illumina MiSeq using gene-specific primers with adapters (*Table 2*) according to the manufacturer's protocol.

## Molecular cloning

GFP was fused to the Dmist_*Dr* ORF by Gateway cloning (*Kwan et al., 2007*). Gene-specific primers were designed to amplify a PCR product that was recombined with middle donor vector (*Table 2*; Invitrogen Gateway pDONR221 Cat# 12536017, Invitrogen Gateway BP Clonase II Cat# 11789020) to generate a middle entry clone (pME-Dmist). pME-Dmist was recombined with 5' (p5E-CMV/SP6) and 3' (p3E-GFPpA) entry clones and destination vector (pDestTol2pA2) using Gateway Technology (Invitrogen LR Clonase II Plus enzyme Cat# 12538200) following the manufacturer's protocol.

A 3 bp mutation was introduced into the *CMV:dreammist-GFPpA* by inverse PCR using specific primers (*Table 2*) and KOD high-fidelity hot start polymerase (Millipore 71085). The template was degraded by DpnI digest and circular PCR product was transformed into OneShot TOP10 chemically competent *Escherichia coli* (Invitrogen C4040). Both *CMV:dreammist-GFPpA* and *CMV:dreammistA22W-GFPpA* constructs were checked by Sanger sequencing.

For labelling the plasma membrane, mRNA was in vitro transcribed from pCS2-myr-Cherry linearised with NotI, in vitro transcribed with SP6 mMessage mMachine (Ambion AM1340), purified and quantified with a QuBit spectrophotometer, and injected at 0.04 µg/µl.

## Microinjection and imaging

For Dmist-GFP and DmistA22W-GFP live imaging, embryos from an AB-TL in-cross were injected with 1 nl of plasmid (7 ng/µl). After developing to 90% epiboly, the embryos were placed on a glass coverslip and observed on an inverted confocal microscope (SPinv, Leica) with a ×40 objective.

## RNAseq

Larvae from heterozygous in-crosses ($dmist^{i8/+} \times dmist^{i8/+}$ and $dmist^{vir/+} \times dmist^{vir/+}$) were raised to adulthood, genotyped, and then homozygous mutant and wild-type siblings were kept separate. Homozygous mutant and wild-type sibling fish were then in-crossed so that first cousins were directly compared. RNA was extracted from thirty 6 dpf larvae using the same protocol as for RT-qPCR and sent for RNAseq analysis at the UCL Institute of Child Health with a sequencing depth of 75 million reads per sample. Differential analysis of transcript count level between groups was performed as in *Love et al., 2014*, and additional analysis was performed using custom MATLAB scripts (MATLAB v9.2 2017, The MathWorks 2017).

## Mouse RNAseq analysis

The dataset was downloaded from https://www.ncbi.nlm.nih.gov/geo/query/acc.cgi?acc=GSE52564 (*Zhang et al., 2014*), and hierarchical clustering (average linkage) and Pearson correlation calculation analysis were performed using custom MATLAB scripts (MATLAB v9.2 2017, The MathWorks 2017).

## Experimental design and statistical analyses

Data was tested for normality using the Kolmogorov–Smirnov test. If data were normally distributed, N-way ANOVA (alpha = 0.05) was used with correction for multiple comparisons using Tukey's test. If non-parametric, the Kruskal–Wallis test was used with correction for multiple comparisons using Dunn–Sidak (alpha = 0.05). Outliers were removed by Grubb's test (threshold p<0.01). p-Values from the LME models were determined by an *F*-test on the fixed effects coefficients generated from the LME model in MATLAB. Data were grouped by genotype and gender for adult experiments and grouped by genotype and day of experiment for larval experiments. The genotype identity of animals was not known prior to tracking and analysis (randomised, blinded), and image analysis was done without knowledge of the group/condition (blinded).

## Acknowledgements

The initial screen, discovery, and characterisation of *dreammist* were conducted in the lab of Alexander F Schier at Harvard University. We also would like to thank members of the Rihel lab and other UCL zebrafish groups for helpful comments on experiments and the article. We thank Shannon Shibata-Germanos for *fxyd1* mutant tracking experiments, John Parnavalas for reagents, Christine Orengo for help with small peptide sequence searches, Stuart Peirson for early access to mouse transcriptomic data, and Finn Mango Bamber for the Pokémon-card-inspired *dreammist* name. The work was funded by NIH grants awarded to Alexander Schier (GM085357 and HL10952505); an ERC Starting Grant (#282027) and Wellcome Trust Investigator Award (#217150/Z/19/Z) to JR; NIH grant R35 NS122172 to DAP; and a Grand Challenges PhD studentship to ILB.

## Additional information

### Funding

| Funder | Grant reference number | Author |
| --- | --- | --- |
| Wellcome Trust | 217150/Z/19/Z | Jason Rihel |
| European Research Council | 282027 | Jason Rihel |
| National Institutes of Health | R35 NS122172 | David A Prober |

| Funder | Grant reference number | Author |
|--------|------------------------|--------|

The funders had no role in study design, data collection and interpretation, or the decision to submit the work for publication. For the purpose of Open Access, the authors have applied a CC BY public copyright license to any Author Accepted Manuscript version arising from this submission.

## Author contributions

Ida L Barlow, Conceptualization, Resources, Data curation, Software, Formal analysis, Supervision, Validation, Investigation, Visualization, Methodology, Writing – original draft, Writing – review and editing; Eirinn Mackay, Resources, Data curation, Formal analysis; Emily Wheater, Data curation, Formal analysis, Investigation, Methodology; Aimee Goel, Formal analysis, Investigation; Sumi Lim, Data curation, Formal analysis, Validation, Investigation, Methodology, Writing – review and editing; Steve Zimmerman, Resources, Data curation, Investigation, Methodology; Ian Woods, Data curation, Formal analysis, Investigation, Methodology, Writing – review and editing; David A Prober, Conceptualization, Resources, Data curation, Formal analysis, Investigation, Methodology, Writing – original draft, Writing – review and editing; Jason Rihel, Conceptualization, Resources, Data curation, Software, Formal analysis, Supervision, Funding acquisition, Validation, Investigation, Visualization, Methodology, Writing – original draft, Project administration, Writing – review and editing

## Author ORCIDs
David A Prober ⓘ http://orcid.org/0000-0002-7371-4675
Jason Rihel ⓘ http://orcid.org/0000-0003-4067-2066

## Ethics

Raising of genetically altered zebrafish and all experimental procedures were performed under project licence 70/7612 and PA8D4D0E5 awarded to JR under the UK Animals (Scientific Procedures) Act 1986 guidelines.

Reviewer #1 (Public Review): https://doi.org/10.7554/eLife.87521.3.sa1
Reviewer #2 (Public Review): https://doi.org/10.7554/eLife.87521.3.sa2
Author Response https://doi.org/10.7554/eLife.87521.3.sa3

# Additional files

## Supplementary files
• MDAR checklist

## Data availability

Data on gene selection is included in *Figure 1—source data 1*. Software used to analyze data are available at https://github.com/ilbarlow/Dmist (copy archived at *Barlow, 2020*) and https://github.com/JRihel/Sleep-Analysis/tree/Sleep-Analysis-Code (copy archived at *Rihel, 2023*). All fish lines are listed in *Table 1* and sperm are frozen at UCL, available upon request. Primers used in this study are listed in *Table 2*. Tracking data used to generate the figures in this article are available at https://github.com/ilbarlow/Dmist.

The following previously published dataset was used:

| Author(s) | Year | Dataset title | Dataset URL | Database and Identifier |
|-----------|------|---------------|-------------|-------------------------|
| Zhang Y, Chen K, Sloan SA, Scholze AR, Caneda C, Ruderisch N, Deng S, Daneman R, Barres BA, JQ Wu | 2014 | An RNA-Seq transcriptome and splicing database of neurons, glia, and vascular cells of the cerebral cortex | https://www.ncbi.nlm.nih.gov/geo/query/acc.cgi?acc=GSE52564 | NCBI Gene Expression Omnibus, GSE52564 |

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
