## [Editor Report · eLife assessment]

This study offers new **fundamental** information on a role for the sodium/potassium pump in sleep regulation. Elegant methods were used to provide **compelling** evidence supporting the claim. The work will be of interest to sleep researchers in zebrafish as well as in other species for future investigation.

---

## [Referee Report · Reviewer #1 (Public Review)]

Barlow et al performed a viral insertion screen in larval zebrafish for sleep mutants. They identify a mutant named dreammist (dmist) that displayed defects in sleep, namely, decreased sleep both day and night, accompanied by increased activity. They find that dmist encodes a previously uncharacterized single-pass transmembrane protein that shows structural similarity to Fxyd1, a Na+K+-ATPase regulator. Disruption of fxyd1 or atp1a3a, a Na+,K+-ATPase alpha-3 subunit, decreased night-time sleep. By staining for sodium levels, the authors uncover a global increase of sodium in both dmist and atp1a3a mutants following PTZ treatment, consistent with defects in Na+K+-ATPase function. These genetic data from multiple mutant lines help establish the importance of sodium and/or potassium homeostasis in sleep regulation.

The conclusions of this paper are mostly well supported by data, with the following strengths and weaknesses as described below.

Strengths:

Elegant use of CRISPR knockout methods to disrupt multiple genes that help establish the importance of regulating Na+K+-ATPase function in sleep.

Data are mostly clearly presented.

Double mutant analysis of dmist and atp1a3a help establish an epistatic relationship between these proteins.

Weaknesses:

The authors emphasize the role of increased cellular sodium, but equally plausibly, the phenotypes could be due to decreased cellular potassium. The potassium channel shaker has been previously identified as a critical sleep regulator in *Drosophila*.

Although the increased sleep rebound after PTZ treatment in the dmist mutant is interesting, I find it difficult to understand, especially in the context of the dmist mutant having decreased sleep.

The similar phenotype between dmist and Fxyd1 in sleep reduction yet very different expression patterns, with dmist being mostly neuronal while fxyd1 being mostly non-neuronal, raise many possible questions: (1) are the sleep phenotypes due to neuronal Na/K imbalance? Or (2) Are the sleep phenotypes due to extracellular Na/K imbalance? Or (3) both? Some feasible experiments may help achieve a better mechanistic understanding of the observed sleep defects.

---

## [Referee Report · Reviewer #2 (Public Review)]

Barlow and colleagues describe a role for the Na+/K+ pump in sleep/wake regulation. They discovered this role starting with a forward genetic screen in which they tested a biased sample of virus insertion fish lines for sleep phenotypes. They found an insertion in a gene they named dreammist, which is homologous to the gene FXYD1 encoding single membrane-pass modifiers of Na/K pumps. They go on to show that genetic manipulations of either FXYD1 or the Na/K pump also reduce sleep. They use pharmacology and sleep deprivation experiments to provide further evidence that the NA/K pump regulates intracellular sodium and rebound sleep. This study provides additional evidence for the important role of membrane excitability in sleep regulation (prior studies have implicated K+ channel subunits as well as a sodium leak ion channel).

The study is well done and convincing with regard to its major conclusions. I had some minor comments/questions, which they properly addressed in their revision and rebuttal.

---

## [Author Response]

The following is the authors’ response to the original reviews.

We thank the reviewers for their comments. We have now addressed all the comments in a revised version of the manuscript, which we believe has strengthened our paper.

1. Introduction LINE 60: the authors cite Funato et al 2016 as the paper first describing a role for SIk3 in sleep regulation. In fact, the role for this kinase was first identified nearly a decade earlier in *C. elegans* (Van der Linden et al, Genetics 2008 PMID 18832350).

Thank you for pointing us to this reference. Van der Linden et al. demonstrated that the *C. elegans* homolog of Sik3 (KIN-29) regulates satiety quiescence, in which worms stop moving following feeding on high quality food. However, as pointed out in Trojanowski and Raizen “Call it Worm Sleep” (2016), not all of the behavioral criteria for sleep has been applied to C. elegans satiety quiescence, and we cannot find any references that unequivocally demonstrate satiety quiescence is a sleep state. As McClanahan et al., (2020) show, quiescent states following mild sensory arousal do not fulfill the sleep criteria of changes in arousal threshold and homeostatic regulation, so not all quiescent states in C. elegans are sleep. Then again Grubbs et al, 2020 does demonstrate that KIN29 regulates both developmentally timed and stress induced sleep states in worms, suggesting that the observations in Van der Linden were ahead of its time and these behavioral states are possibly inter-related. We believe, though, that our line “the roles of… SIK3 kinase in modulating sleep homeostasis in mice (Funato et al. 2016) were identified in genetic screens” remains accurate.

1. Introduction LINE 71: remove the word "known" from "...while some known human sleep/wake regulators, such as the..."

Good idea. Done.

1. I was confused regarding Supplemental data 1 describing the genes they targeted with their forward genetic screen. Am I understanding correctly from the "Summary stats" tab that 702 fish lines with virus insertions were screened behaviorally? In Figure S1, it looks like about 60 are shown in the histograms but in the text (in the Discussion) they say 25 were screened. Were all the genes listed under the Excel tabs (GPCRs, channels, etc) tested? Or was just a subset tested? Where are the sleep data for these lines? Negative results may be relevant to their manuscript since they listed (tested??) a number of ion channel genes under tab "channels" which appear to NOT have a sleep phenotype.

We apologize for the confusion on these points. As highlighted in the legend to Supplementary Figure S1, we had planned a screening strategy with the following pipeline: Candidate mammalian gene → Zebrafish ortholog → ID viral insertion from “Zenemark” library → grow viral insertion lines from frozen sperm→ phenotype F3 heterozygous and homozygous mutant generation. Unfortunately, the company, Znomics, which held the Zenemark library, could not reliably reconstitute the correct live fish from the sperm library, and of the 702 lines we planned to screen, we could only screen 26 (25 was a typo) lines. We treated heterozygous and homozygous animals for each line independently, for a total of 52 screened lines in the histograms.

To make this clearer, we have edited the main text as follows (lines 104-105): “For screening, we identified zebrafish sperm samples from the Zenemark collection (Varshney et al., 2013) that harboured viral insertions in genes of interest and used these samples for in vitro fertilization and the establishment of F2 families, which we were able to obtain for 26 lines.” And lines 111-112: “While most screened heterozygous and homozygous lines had minimal effects on sleep-wake behavioural parameters (Figure S1B-S1C),”

We believe it is important to include the full set of Supplementary Data 1, even though the vast majority of these candidate lines were not tested.

1. Results LINE 117: remove the word "prominent", which is subjective, from the sentence "...showed a prominent decrease in sleep during the..."

Good point. Done.

1. LINES 185-186: did you see any circadian variation in your dmist:GFP protein abundance or localization? Protein trafficking has been described as a mechanism of circadian regulation of excitability.

For practical reasons, we imaged the membrane localization of Dmist:GFP in plasmidinjected embryos at 90% epiboly, which is about 9 hours after fertilization and when the cells remain large and in a relatively flat epithelium. Thus, we could not follow circadian fluctuations in abundance or localization. For circadian studies, we believe the best method will be to raise an antibody that recognizes Dmist.

1. LINE 203: does the GFP-tagged Dmist rescue the loss-of-function phenotype? This is relevant to Figure 2E. it is also relevant to the issue of structure-function. If it rescues, then the C-terminus may not be essential to protein function.

As noted, for practical reasons, we observed Dmist-GFP only transiently at early stages of development, expressed using a strong, ubiquitous promoter. A rescue experiment is a good idea for future experiments, where we carefully control the expression of Dmist in neurons.

1. LINE 220: explain what you mean by "...consistent with nonsense-mediated decay." and/or give a reference.

In zebrafish and other species including humans, mutant transcripts that have premature stop codons often undergo “nonsense mediated decay”, whereby the expression levels are largely reduced (Wittkopp et al., 2009). In the zebrafish community, this is often used as secondary evidence of a loss of function mutation, as relatively few antibodies are available to directly observe zebrafish proteins. We have added a reference that describes this phenomenon (Wittkopp et al., 2009).

1. LINE 225: define "LME model"

Now reads: “Linear mixed effects (LME).”

1. LINES 227-229: could the vir/vir phenotype be explained by specific effects on protein structure? could vir/vir be a gain-of-function allele?

We can’t rule this out formally, and vir/+ animals do show some sleep phenotypes, albeit weaker than those of vir/vir animals (Figure 1G). However, it is not uncommon for heterozygous mutants to show significant phenotypes that are weaker than those of their homozygous mutant siblings, and the strong suppression of dmist expression by the viral insertion (which is located in the dmist intron) is more consistent with a hypomorphic loss-of-function phenotype for the vir allele.

1. LINES 229-230: I don't quite follow the argument for pursuing further studies only of i8/i8. i8/i8 seems to also be a hypomorphic allele based on your qPCR data.

First, the dmist viral line was generated by an insertional mutagenesis method followed by sequencing, and each line has multiple other inserts in a background that does not match the background of the other animals reported in this paper. Second, the dmist vir allele is an insertion in the intron, leading to reduced, but not complete loss of expression. In contrast, the i8 allele was generated on the same background strain as our other existing and newly reported lines. Moreover, our i8 line is likely a loss-of-function allele and not a hypomorph. Yes, dmist expression is reduced in the i8 allele; however, this is likely due to nonsense mediated decay of dmist mRNA. The mutation introduces a frameshift in the dmist coding sequence, and as a result the amino acid sequence of the protein is altered after the N-terminal signal sequence.

1. LINES 241-243: grammar.

Fixed

1. LINE 245: define "JackHMMR iterative search"

We’ve added the phrase: “and seeding a hidden Markov model iterative search (JackHMMR)”

1. LINE 246 is missing the word "we" prior to "...found distant homology between..."

Added

1. LINE 301: show data demonstrating deviation from Mendelian ratios. Also, comment on meaning of such data (embryonic lethality??).

We have added this data in the line (301):

“atp1a3b mutant larvae were not obtained at Mendelian ratios (55 wild type [52.5 expected], 142 [105] atp1a3b+/-, 13 [52.5] atp1a3b-/-; p<0.0001, Chi-squared) suggesting some impact on early stages of development leading to lethality.”

1. Discussion LINES 362-372: This paragraph seems to be of only tangential relevance to the paper. Consider removing.

Our screening strategy was a large-scale reverse genetic screen, but the number of lines was limited by the technical issues described above. We think it is important to mention that the strategy, if employed today, could benefit from newer technologies.

1. Discussion. Another model is that Dmist and NaK pump have a developmental effect. Arguing against this developmental model is the Oubain expt.

This is an important point. We’ve added the line (454:457): “We also cannot exclude a role for Dmist and the Na+/K+ pump in developmental events that impact sleep, although our observation that ouabain treatment, which inhibits the pump acutely after early development is complete, also impacts sleep, argues against a developmental role.”

1. FIGURE 1G: Are these significance cut offs corrected for multiple comparisons?

Yes, all the data is corrected for multiple comparisons.

1. performing neuronal activity measures, either via neural activity imaging or phospho-ERK labeling in different mutants at day or night conditions, to determine whether baseline neuronal activity brain-wide or in specific brain regions are altered.

These are excellent experiments that we plan to perform in the future.

1. Please check all Figure numbers for accuracy.

We have double checked these.

1. The authors emphasize the role of increased cellular sodium, but equally plausibly, the phenotypes could be due to decreased cellular potassium. The potassium channel shaker has been previously identified as a critical sleep regulator in *Drosophila*.

We completely agree. We would like to highlight that we did devote an entire paragraph to the possibility of changes in extracellular potassium in the discussion: “A third possibility is that Dmist and the Na+,K+-ATPase regulate sleep not by modulation of neuronal activity per se but rather via modulation of extracellular ion concentrations. Recent work has demonstrated that interstitial ions fluctuate across the sleep/wake cycle in mice. For example, extracellular K+ is high during wakefulness, and cerebrospinal fluid containing the ion concentrations found during wakefulness directly applied to the brain can locally shift neuronal activity into wake-like states (Ding et al., 2016). Given that the Na+,K+-ATPase actively exchanges Na+ ions for K+ , the high intracellular Na+ levels we observe in atp1a3a and dmist mutants is likely accompanied by high extracellular K+. Although we can only speculate at this time, a model in which extracellular ions that accumulate during wakefulness and then directly signal onto sleep-regulatory neurons could provide a direct link between Na+,K+ ATPase activity, neuronal firing, and sleep homeostasis. Such a model could also explain why disruption of fxyd1 in non-neuronal cells also leads to a reduction in night-time sleep.”

We also agree that Shaker may be an important component of this sleep regulatory mechanism. Indeed, we previously showed that another potassium channel in zebrafish regulates sleep (Rihel et al., 2010).

We have emphasized sodium homeostasis in our title and paper only because we were able to directly observe intracellular sodium levels, so we are confident that these have been altered in our mutants. We can only presume that potassium levels have also been altered, but we could not directly observe this.

1. The similar phenotype between dmist and Fxyd1 in sleep reduction yet very different expression patterns, with dmist being mostly neuronal while fxyd1 being mostly non-neuronal, raise many possible questions: (1) are the sleep phenotypes due to neuronal Na/K imbalance? Or (2) Are the sleep phenotypes due to extracellular Na/K imbalance? Or (3) both? Some feasible experiments may help achieve a better mechanistic understanding of the observed sleep defects.

Yes, we think these are excellent studies for future work. As noted in the previous point (20), we did discuss the possibility that changes to extracellular potassium might be a parsimonious explanation for the similar phenotypes of fxyd1 and dmist mutants.

Future experiment suggestions (not required)1. Perform a double mutant analysis of fxyd1 and atp1a3a, to determine whether an epistatic relationship similar to that of dmist and atp1a3a is observed in the case of fxyd1 and atp1a3a.

This is a great experiment that we will do in the future. Unfortunately, the fxyd1 mutant had been sperm frozen during the COVID-19 pandemic, so we cannot do this experiment at this time.

1. Given the differences in the sleep phenotypes between vir/vir and i8/i8 mutants, would be informative to see the phenotype of the vir/i8 trans-heterozygote.

This is also a good experiment to perform in the future. Since obtaining the cleaner i8 allele, the dmistvir/vir lines were sperm frozen.